# LFA-1 nanoclusters integrate TCR stimulation strength to tune T-cell cytotoxic activity

Claire Lacouture [1,2], Beatriz Chaves[1,3,4], Delphine Guipouy[1], Raïssa Houmadi [1], Valérie Duplan-Eche [1], Sophie Allart[1], Nicolas Destainville [2] ✉ & Loïc Dupré [1,5] ✉

T-cell cytotoxic function relies on the cooperation between the highly specific but poorly adhesive T-cell receptor (TCR) and the integrin LFA-1. How LFA-1-mediated adhesion may scale with TCR stimulation strength is ill-defined. Here, we show that LFA-1 conformation activation scales with TCR stimulation to calibrate human T-cell cytotoxicity. Super-resolution microscopy analysis reveals that >1000 LFA-1 nanoclusters provide a discretized platform at the immunological synapse to translate TCR engagement and density of the LFA-1 ligand ICAM-1 into graded adhesion. Indeed, the number of high-affinity conformation LFA-1 nanoclusters increases as a function of TCR triggering strength. Blockade of LFA-1 conformational activation impairs adhesion to target cells and killing. However, it occurs at a lower TCR stimulation threshold than lytic granule exocytosis implying that it licenses, rather than directly controls, the killing decision. We conclude that the organization of LFA-1 into nanoclusters provides a calibrated system to adjust T-cell killing to the antigen stimulation strength.

A pivotal question of biology is to understand how cells integrate stimulatory triggers into calibrated tasks. Upon stimuli sensing, cells activate a series of biochemical and biophysical events to execute specialized tasks such as polarized secretion. Exploring the metrics relating stimuli sensing to downstream events at a single-cell resolution is challenging but necessary to understand how cells and cell populations deliver calibrated tasks. Task calibration is particularly central for the operation of the immune system, whereby highly specialized cells constantly adapt to environmental cues to deliver graded responses[1–3]. In particular cytotoxic T cells must finely tune their cytotoxic activity to eliminate infected cells and tumor cells while sparing surrounding healthy cells[4,5]. Understanding cytotoxic activity calibration is key to not only characterize physiological responses but also to optimize therapeutic T-cell strategies[6].

Antigen-dependent activation via the TCR and adhesion to target cells via LFA-1 represent the minimal set of signals that trigger the commitment of CD8[+] T cells to kill target cells. Remarkably, those signals are related in a cooperative fashion since cognate TCR engagement promotes the conformation-dependent activation of LFA-1 to stabilize the interaction of T cells to ICAM-1 expressing target cells and thereby ensure the sustainment of TCR engagement[7–11]. Multiple lines of evidence highlight the pivotal role of adjusting TCR-driven activation of LFA-1 to regulate CD8[+] T-cell cytotoxic function. Efficacy and strategy of killing appears to be determined by the timing of adhesion to target cells[12]. The assembly of a high-affinity LFA-1 belt has been proposed to allow confined delivery of lytic granule content[13]. Furthermore, forces engaged through LFA-1 have recently been shown to sustain local delivery of lytic granules[14]. However, the quantification of the TCR-LFA-1 cooperation and its translation into adhesion, lytic granule secretion and cytotoxic potency have remained elusive.

The prevailing model of LFA-1 activation at the spatial level has been that TCR-evoked affinity maturation is accompanied by an active

[1]Toulouse Institute for Infectious and Inflammatory Diseases (INFINITy), INSERM, CNRS, Toulouse III Paul Sabatier University, Toulouse, France. [2]Laboratoire de Physique Théorique, Université de Toulouse, CNRS, UPS, Toulouse, France. [3]National Institute of Science and Technology on Neuroimmunomodulation (INCT-NIM), Oswaldo Cruz Institute, Oswaldo Cruz Foundation (Fiocruz), Rio de Janeiro, Brazil. [4]Computational Modeling Group, Oswaldo Cruz Foundation (Fiocruz), Eusébio, Brazil. [5]Department of Dermatology, Medical University of Vienna, Vienna, Austria. ✉e-mail: destain@irsamc.ups-tlse.fr; loic.dupre@inserm.fr

process of LFA-1 microcluster assembly[15–18]. However, super-resolution microscopy data from our group and others have revealed that, even before activation, LFA-1 organizes as discrete nanoclusters rather than larger microclusters as previously assumed[19–23]. How distinct LFA-1 conformations distribute in nanoclusters has not yet been investigated and would be required to better grasp the process of affinity maturation. Quantifying LFA-1 conformations at a nanoscale resolution would also help understanding how TCR activation strength might calibrate adhesion and associated cytotoxic activity efficacy. Given the variable levels of ICAM-1 densities on target cells, and its key role in cell-cell adhesion, understanding the impact of ICAM-1 density on the efficacy of the TCR-evoked inside-out signaling would also be crucial to predict efficacy of cytotoxic T cells in clinical settings[24]. Finally, specifically blocking the conformational activation would be required to directly test whether it is required to mediate the cytotoxic activity.

In this work, we assessed how a graded stimulation of the TCR translates into LFA-1 conformation activation by focusing on the low-affinity and high-affinity conformations among the spectrum of conformation intermediates described for LFA-1[10,25]. We characterize LFA-1 activation at the nanoscale and the cellular scale, and explore how LFA-1 activation relates to interaction with target cells, secretion of lytic granules and cytotoxicity. We show that LFA-1 conformations segregate in distinct nanoclusters and that the number of open conformation nanoclusters scales with TCR stimulation strength. We identify that LFA-1 conformational activation licenses CD8[+] T cells to degranulate and kill, but that adhesion time does not act as a limiting factor for killing. Our data also highlight the impact of ICAM-1 availability on LFA-1 activation and killing. Together our findings highlight how a crucial biological activity is robustly controlled via the graded activation of cellular adhesion sustained by discrete nanoclusters.

## Results

### LFA-1 conformations segregate in distinct nanoclusters at the CD8[+] T cell synapse

To gain insight into the activation of LFA-1 in the context of the human CD8[+] T cell synapse, we first aimed at resolving the spatial distribution of the closed and open conformations of LFA-1 at the nanoscale. For that purpose, CD8[+] T cells were stimulated with coated ICAM-1 and anti-CD3 Ab and co-stained with the Hi-111 and m24 antibodies (Ab) specific of the low-affinity and high-affinity conformations of LFA-1, respectively[26,27]. Cell imaging with Stimulated Emission Depletion (STED) microscopy revealed that the two LFA-1 conformations distributed in a segregated fashion, each with high local density fluctuations. Indeed each conformation appeared to distribute into distinct discrete clusters with a typical diameter of 100 nm (Fig. 1a). A machine learning approach that we validated with artificial images composed of randomly distributed molecules and clusters (see "Methods" and Supplementary Material) was applied to the STED images in order to automatically detect nanoclusters (Fig. 1b). It revealed that, although the activated nanoclusters represent only 6% of the total surface area, they contain about 20% of the m24 staining total staining (the ratio for inactivated clusters detected with Hi-111 staining is 10% of the surface for 18% of the intensity) (Supplementary Fig. 1a). This suggests that a substantial fraction of the LFA-1 molecules are organized in nanoclusters, the remaining fraction corresponding to a diffuse background that cannot be resolved by STED microscopy. Most detected clusters were single-conformation nanoclusters, including closed conformation nanoclusters (area composed by more than 95% of Hi-111 Ab staining) and open conformation nanoclusters (area composed by more than 95% of m24 Ab staining). Fewer objects detected as mixed clusters and presenting a combination of closed and open conformations (area composed by both Hi-111 and m24 Abs stainings with more than 5% overlap) were detected (Fig. 1c and Supplementary Fig. 1b). Analysis of cluster area distribution indicated that closed and open conformation nanoclusters harbored a typical size,

with comparable median surfaces of 6800 and 5400 nm², corresponding to diameters of about 90 and 80 nm, respectively (Fig. 1d). The objects detected as mixed clusters by STED displayed a broader area distribution and a median size equivalent to two typical single-conformation nanoclusters or more. To further examine the nature of the mixed clusters, the proportion of conformation overlap was analyzed (Fig. 1e). Mixed clusters appeared to harbor only limited overlap of LFA-1 conformations (median overlap of 20%) and limited intermixing of the two conformations. Therefore, the objects detected as mixed clusters likely correspond to the juxtaposition of closed and open nanoclusters that could not be resolved by STED imaging (Fig. 1f). The distribution of LFA-1 conformations into nanoclusters of equivalent sizes was confirmed by independent dSTORM analysis (Supplementary Fig. 1c, d). Further analysis of dually stained cells by SIM also confirmed the segregation of closed and open conformations of LFA-1 into distinct nanoclusters (Supplementary Fig. 1e, f). Taken together, these data show that at least parts of the closed and open conformations of LFA-1 distribute in distinct nanoclusters at the synapse of stimulated CD8[+] T cells, which is suggestive of a local coordination of affinity state maturation.

### Number of open conformation LFA-1 nanoclusters scales with TCR stimulation strength

We then explored how the nanoscale distribution of LFA-1 conformations might respond to varying TCR stimulation strength. For that purpose STED imaging of the closed and open LFA-1 conformations was further applied to CD8[+] T cells that were either unstimulated, stimulated with ICAM-1 alone or stimulated with ICAM-1 and various anti-CD3 Ab concentrations (Fig. 2a). In all tested conditions and for all observed cells, LFA-1 nanoclusters were detected, most of which harboring a single conformation. This was also the case for cells stained in suspension, at the surface of which closed conformation nanoclusters were detected (Supplementary Fig. 2a, b). These data indicate that LFA-1 molecules tend to be assembled in nanoclusters independently from stimulation. To then explore how stimulation impacts LFA-1 nanocluster topography, the number of detected nanoclusters was computed across the tested stimulation conditions (Fig. 2b, c and Supplementary Fig. 2c). T cells deposited on Poly-L-Lysine (PLL) harbored in average 400 nanoclusters at the contact plane, the vast majority of which being closed conformation nanoclusters. T cells deposited on ICAM-1 harbored increased numbers of both closed and open conformation nanoclusters, suggesting that engagement of LFA-1 with its ligand promotes the recruitment and the activation of LFA-1 nanoclusters. Upon co-engagement of the TCR with increasing anti-CD3 Ab concentrations, T cells displayed a gradual increase in detected open conformation nanoclusters, while the numbers of closed conformation nanoclusters remained relatively stable. In line with these measurements, the proportion of open conformation nanoclusters was found to increase as a function of anti-CD3 Ab concentration, to reach up to 52% at the 10 μg/ml concentration (Fig. 2c). The number of detected LFA-1 nanoclusters was then normalized to the area of each of the studied cells. The calculated densities of open conformation nanoclusters per cell surface unit increased gradually with the stimulation (Fig. 2d). Our analysis also indicated that the median area of the LFA-1 nanoclusters was not impacted by the stimulation (Supplementary Fig. 2d). We further observed that the densities of detected clusters scaled linearly with the global staining intensities across the stimulation conditions (Supplementary Fig. 2e). Collectively, these data reveal that the process of LFA-1 activation in the context of CD8[+] T cell synapse assembly is sustained by the enrichment of open conformation nanoclusters at the stimulatory area, the number rather than the size of which grows with TCR stimulation strength. Since a fraction of LFA-1 is detected as closed conformation nanoclusters at the surface of unstimulated cells, LFA-1 activation might not only result from a de novo molecular assembly

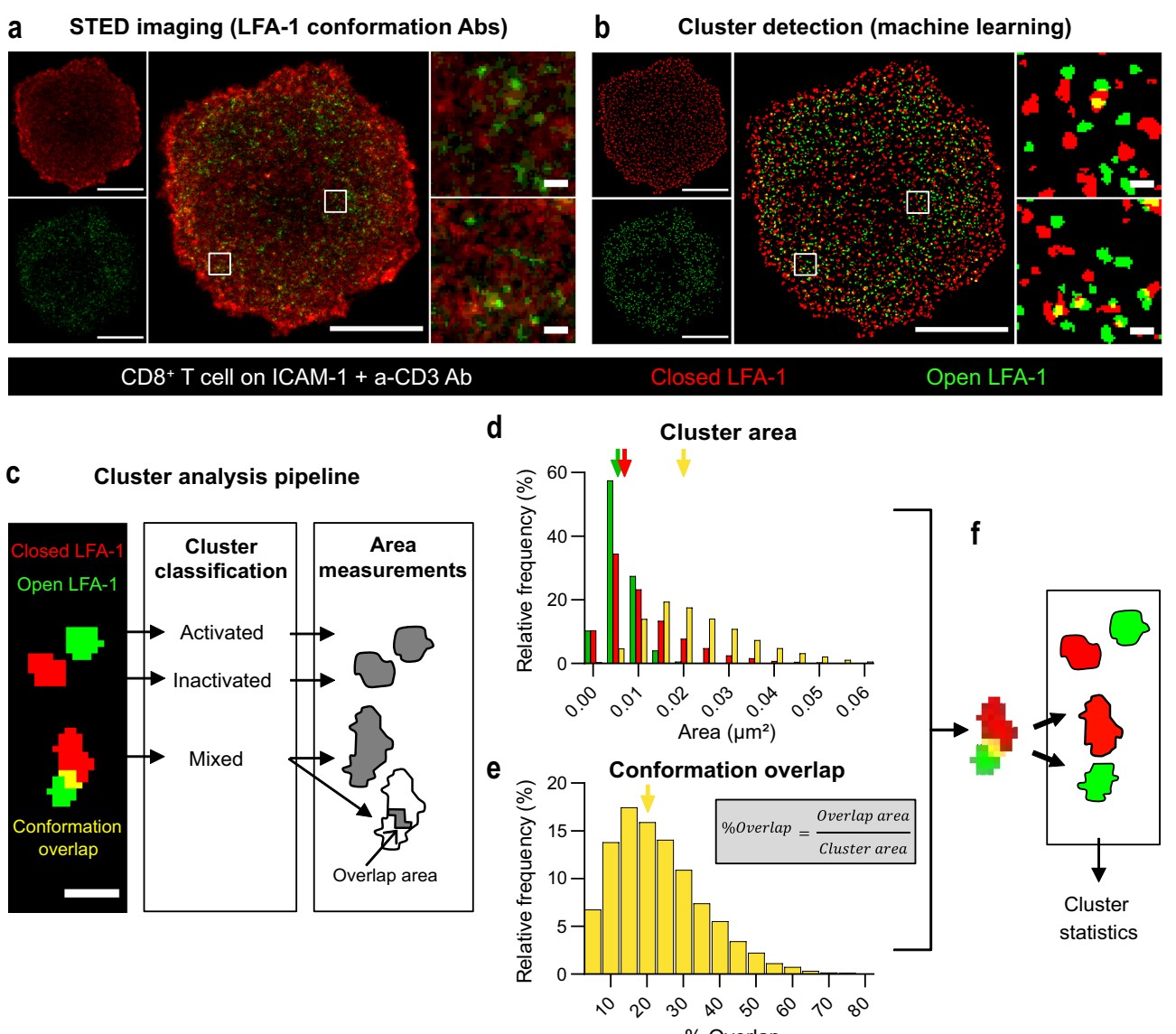

**Fig. 1 | Nanoscale organization of LFA-1 at the CD8+ T cell synapse. a** STED image of the adhesion plane of a representative CD8+ T cell spreading on ICAM-1 + anti-CD3 Ab for 20 min prior to fixation. Closed (red) and open (green) conformations of LFA-1, as revealed with Hi-111 and m24 Ab, respectively. Image from one experiment out of two independent experiments. Scale bars: 5 μm (whole cell) and 0.2 μm (ROI). **b** LFA-1 clusters as detected by the Trainable Weka Segmentation algorithm applied to the STED image presented in (**b**). **c** Representative nanocluster types (scale bar: 150 nm) were further processed to extract nanocluster area (for closed, open and mixed clusters) and conformation overlap area (for mixed clusters). **d** Distribution of nanocluster areas (28 cells studied, from two independent experiments). The arrows point to the median area of the closed (red), open (green) and mixed (yellow) clusters, which are, respectively, 0.0054, 0.0068 and 0.0201 μm². **e** Distribution of the percentage of conformation overlap in the nanoclusters detected as mixed (28 cells studied, from two independent experiments). The arrow indicates the median value (20%). **f** Schematic representation of the treatment of mixed cluster events: since they display limited conformation overlap and their size exceeds that of single-conformation nanoclusters, these events are considered as juxtaposed nanoclusters. Source data are provided as a Source Data file.

process as hypothesized so far but also from a conformation switch of preassembled clusters.

## Activation and distribution of high-affinity LFA-1 in the context of synapse assembly

To further investigate the process of LFA-1 conformational activation as a function of TCR stimulation strength, we turned to a high-throughput confocal imaging pipeline[28] to automatically extract morphological features and intensities of closed and open conformation LFA-1 in a high number of synapse-forming CD8+ T cells. As depicted in representative images, cells deposited on ICAM-1 adopted a highly polarized and elongated morphology, while cells deposited on ICAM-1 and anti-CD3 Ab displayed a radial spreading (Fig. 3a).

Increasing concentrations of anti-CD3 Ab appeared to promote a more circular shape, as well as a relocation of open conformation LFA-1 from one pole of the cell to a belt-like structure. In agreement with the STED data, increasing TCR stimulation strength also resulted in the gradual increase of open conformation LFA-1 intensities, while those of closed conformation LFA-1 remained unaffected (Fig. 3b). The cells displaying a high level of LFA-1 activation corresponded to the cells that also displayed a high level of closed LFA-1 conformation, suggesting that LFA-1 activation scales with the level of available LFA-1 at the surface of individual cells (Fig. 3c). From a morphological point of view, the spreading area was only minimally affected by the presence and concentration of anti-CD3 Ab (Supplementary Fig. 3a) and did not seem to correlate with LFA-1 activation (Fig. 3d). The roundness of the cells was

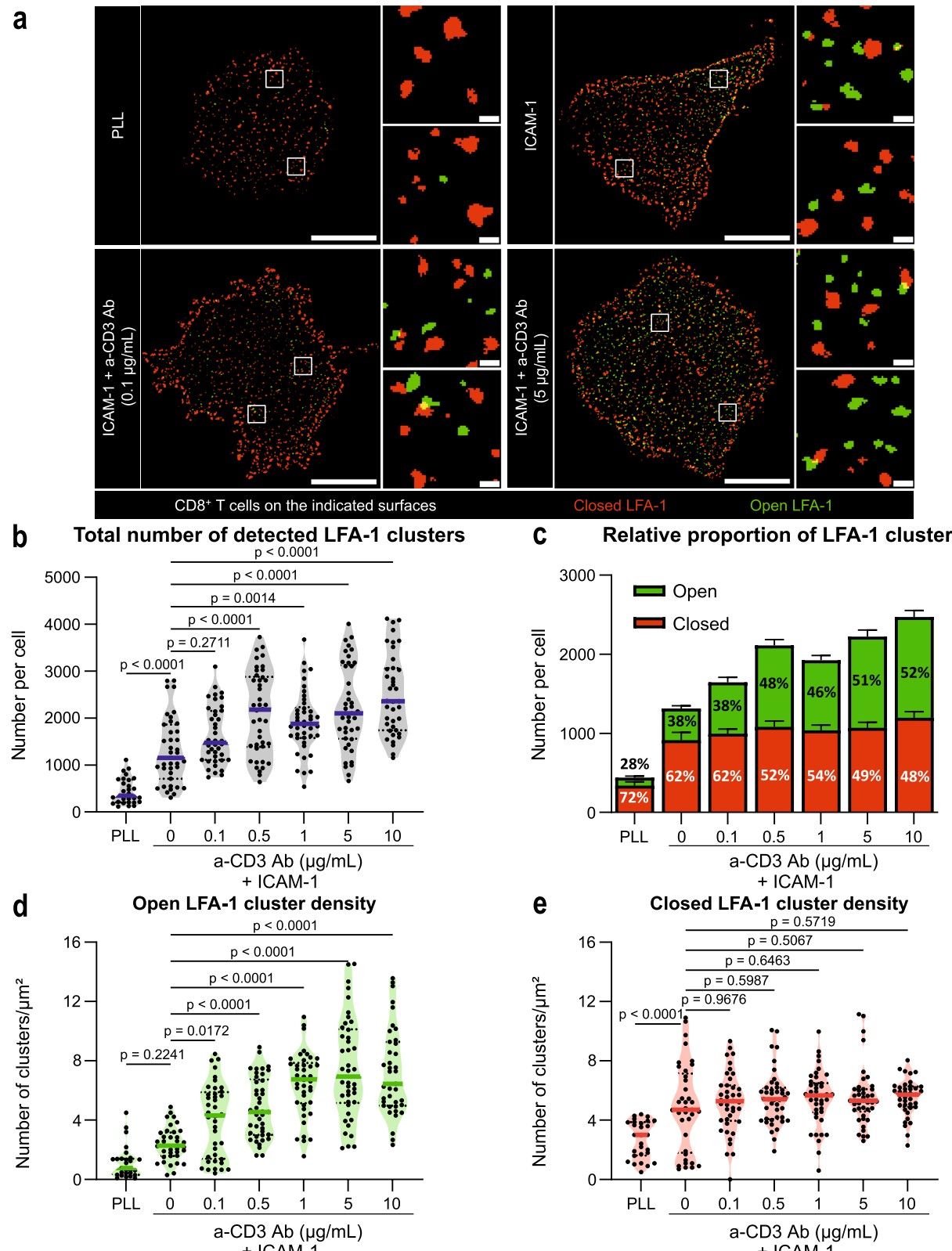

**Fig. 2 | Conformational state of LFA-1 nanoclusters as a function of TCR stimulation. a** LFA-1 nanoclusters as detected by the segmentation algorithm from STED images of CD8[+] T cells spreading over the indicated surfaces. Closed (red) and open (green) LFA-1 conformations, as revealed with Hi-111 and m24 Ab, respectively. Scale bars: 5 μm (whole cell) and 0.2 μm (ROI). **b** Total number of LFA-1 nanoclusters detected at the adhesion plane of T cells spreading over the indicated surfaces. **c** Number and proportion of closed or open clusters. Densities of open (**d**) and closed (**e**) LFA-1 nanoclusters. **b**–**e** Pooled results of three independent experiments conducted on T cells from two donors. In total, 27, 38, 40, 42, 41, 42 and 39 cells were studied, respectively for PLL, 0, 0.1, 0.5, 1, 5 and 10 μg/ml anti-CD3 Ab. Dots: individual cells, lines: median, bars: SEM. *p* values calculated by parametric one-way ANOVA tests. Source data are provided as a Source Data file.

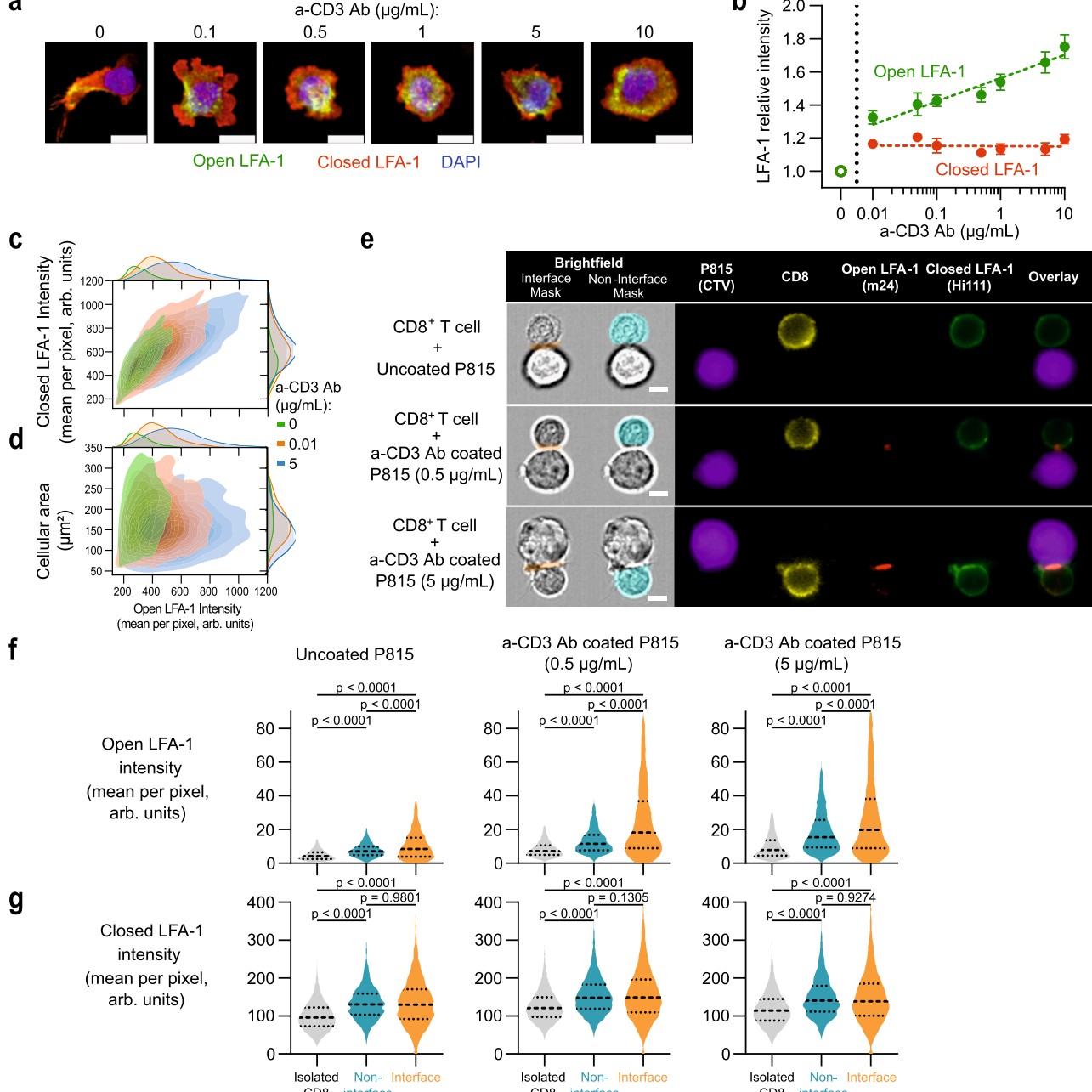

**Fig. 3 | Single-cell analysis of LFA-1 conformation activation at the CD8+ T-cell synapse. a** Representative images of CD8+ T cells spreading on ICAM-1 and the indicated concentrations of a-CD3 Ab. Blue: nucleus, red: closed LFA-1, green: open LFA-1. Scale bar: 10 μm. **b** Relative intensity (compared to ICAM-1 alone) of closed (red) and open (green) LFA-1 of cells adhering to ICAM-1 and the indicated concentrations of anti-CD3 Ab. Each symbol represents the mean and SEM of four independent experiments, each including two replicates. The number of studied cells are 3143, 5694, 7159, 8164, 8143, 8421, 8750 and 9317, respectively for the 0 to 10 μg/ml anti-CD3 Ab conditions. Dotted lines represent linear regression on the mean values type a*log[a-CD3 concentration] + b. The slope is significantly positive for open LFA-1 ($p$ value < 0.0001) but not significantly different to zero for closed LFA-1 ($p$ value: 0.8925). **c** Distribution of closed and open LFA-1 intensities across

individual cells. One representative experiment aggregating two duplicates among four independent experiments is shown. **d** Distribution of cellular area and open LFA-1 intensity across individual cells. One representative experiment aggregating two duplicates among four independent experiments is shown. **e** Representative images of CD8+ T cells in contact with P815 cells coated with the indicated concentrations of anti-CD3 Ab, acquired with a flow imager. Scale bar: 5 μm. Intensity of m24 (**f**) and Hi-111 Ab (**g**) in isolated CD8+ T cells (gray) or CD8+ T cells in contact with P815 cells (blue: non interface, orange: interface). The data of two independent experiments are pooled. The bars represent the first, second (median) and third quartile. $p$ values calculated by parametric one-way ANOVA tests. Gating strategy exposed in Supplementary Fig. 4a. Source data are provided as a Source Data file.

strongly impacted by the presence of anti-CD3 Ab but did not directly scale with LFA-1 conformational activation (Supplementary Fig. 3b, c). This analysis indicates that the rate of LFA-1 conformational activation might not translate linearly into remodeling of adhesive response such as the transition from elongated to radial spreading in the context of

immunological synapse assembly. We then turned to imaging flow cytometry in order to investigate the distribution of closed and open conformations of LFA-1 in the context of contacts between CD8+ T cells and anti-CD3 Ab-coated P815 target cells (see Supplementary Fig. 4a for analytical pipeline). While open-conformation LFA-1 was mostly

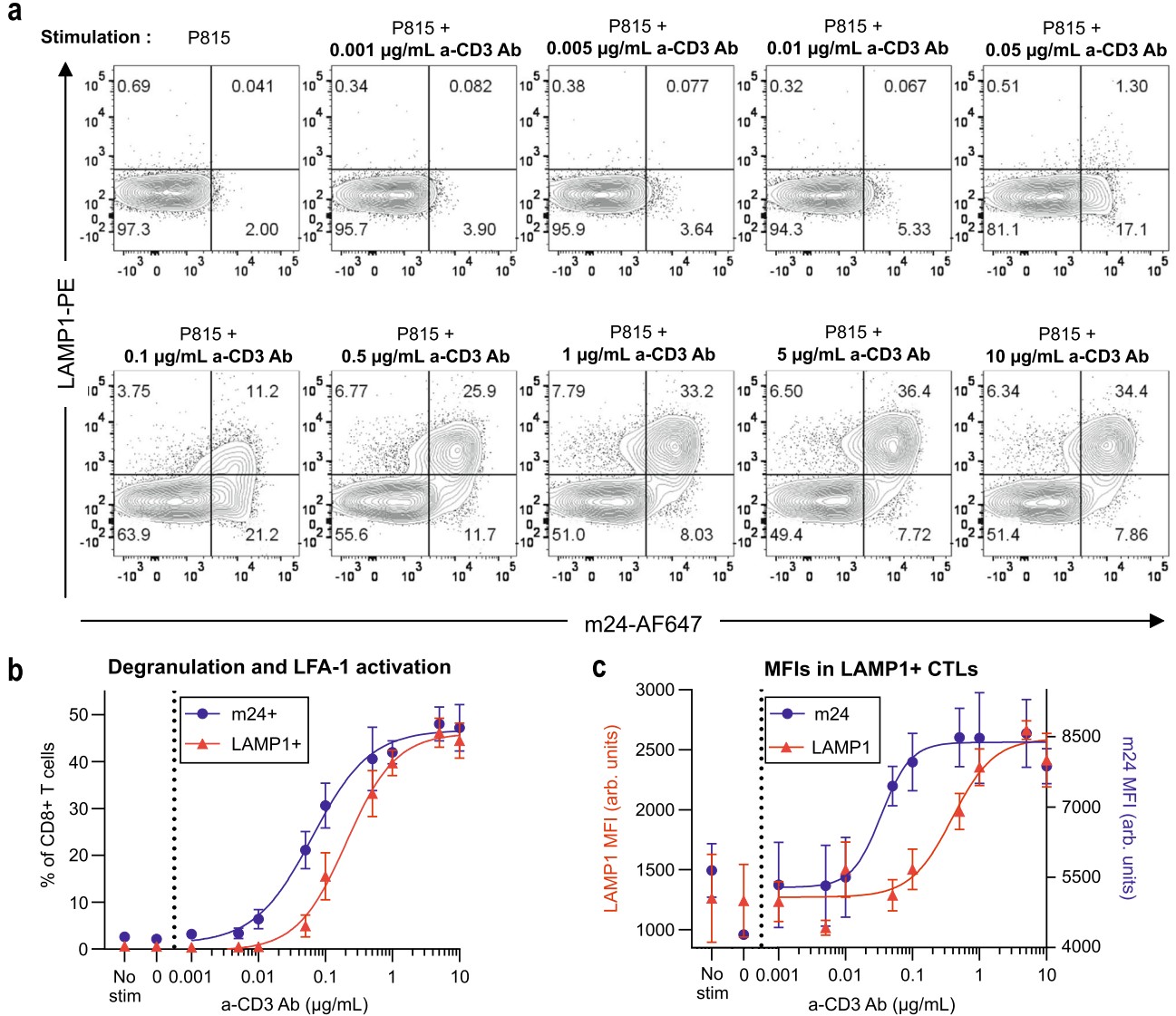

**Fig. 4 | Relationship between LFA-1 activation and lytic granule exocytosis.**
**a** Flow cytometry analysis of LAMP1 and m24 stainings of CD8⁺ T cells stimulated by P815 cells coated with the indicated concentrations of anti-CD3 Ab. One representative experiment among three independent experiments is shown. $N = 10,000$ cells per condition. **b** Percentage of LAMP1⁺ (red) and m24⁺ (blue) CD8⁺ T cells after 2-h stimulation with P815 cells coated with the indicated concentrations of a-CD3 Ab. The dots represent the mean of three independent experiments (each with 30,000 cells per condition) and error bars correspond to SEM. The lines represent sigmoidal fits. Corresponding gates are shown in (**a**). **c** Mean fluorescence intensity (geometric mean, MFI) of LAMP1 (red, left axis) and m24 (blue, right axis) in LAMP1⁺ T cells. The dots represent the mean of three independent experiments (each with 30,000 cells per condition) and error bars correspond to SEM. The lines represent sigmoidal fits. Gating strategy exposed in Supplementary Fig. 5a. Source data are provided as a Source Data file.

confined within the synapse, closed-conformation LFA-1 remained in equilibrium between the synapse and the rest of the cell (Fig. 3e–g and Supplementary Fig. 4b, c). This analysis suggests that open-conformation LFA-1 is trapped within the synapse and that closed-conformation LFA-1 diffuses from the membrane outside the synapse to replenish the synaptic fraction that converts to open conformation, which could explain the fact that we observe a constant closed LFA-1 staining at the immunological synapse in previous figures. Together the high-throughput analysis of LFA-1 activation in the context of 2D synapses and synapses with target cells reinforces the notion that high-affinity maturation scales with TCR stimulation and is confined to the synapse area, presumably through its interaction with ICAM-1.

## LFA-1 conformational switch relates to lytic granule exocytosis

LFA-1 mediated adhesion is a well-established prerequisite for contact-dependent cytotoxicity[13,14,29]. In this context, we next asked how the varying degrees of LFA-1 activation revealed here may relate to key parameters of the cytotoxic activity, such as adhesion to target cells and degranulation. We first assessed the relationship between TCR stimulation, LFA-1 activation and degranulation by measuring LFA-1 conformational activation and LAMP1 surface expression in CD8⁺ T cells by flow cytometry (Fig. 4a and Supplementary Fig. 5a). To this purpose, CD8⁺ T cells were co-cultivated with P815 target cells coated with various anti-CD3 Abs concentrations for 2 h. In response to increasing anti-CD3 Ab concentrations, both the proportion of m24⁺ cells and LAMP1⁺ cells followed a sigmoid function (as well as the percentage of P815:T-cell doublets and the percentage of dead P815 cells, Supplementary Fig. 5b, c). Remarkably however, the activation of LFA-1 was found to occur at a five-fold lower anti-CD3 Ab concentration than that required to trigger LAMP-1 surface exposure, suggesting that cytotoxic T cells do not commit to lytic granule secretion unless they have engaged in tight adhesion with target cells via LFA-1 conformation activation. In agreement with this notion, single cell analysis showed that degranulation occurred almost exclusively in cells that

presented high open conformation LFA-1 levels (Supplementary Fig. 5d). Interestingly, the intensity of the LAMP1 staining within LAMP1⁺ cells also increased with the anti-CD3 Ab concentration, although again with a threshold shift as compared to the proportion of LAMP1⁺ cells (Fig. 4c). This suggests that besides setting the proportion of CD8⁺ T cells that degranulate, TCR stimulation might also tune the rate of secreted granules. Together these data provide quantitative insight into the understanding of how cytotoxic T cells calibrate their adhesive and secretory responses as a function of TCR stimulation.

## TCR stimulation strength determines stability of contacts with target cells and killing

We next investigated how adhesion parameters expected to depend on LFA-1 activation might relate to the onset of the cytotoxic activity itself. For this purpose, we imaged by video-microscopy anti-CD3 Ab coated P815 target cells exposed to CD8⁺ T cells and recorded lethal hits events using propidium iodide (Fig. 5a and Movie 1). Our analysis shows that contacts led to killing only above the anti-CD3 Ab concentration of 0.1 μg/ml (Fig. 5b), which is in agreement with the LAMP1 surface exposure measurements. Remarkably, the contact time measured for each CD8⁺ T cell and target cell conjugate, was largely extended in response to increasing anti-CD3 Ab concentrations (Fig. 5c). Indeed, whereas contacts of CD8⁺ T cells with uncoated target cells or target cells coated with 0.1 μg/ml anti-CD3 Ab lasted for 15 min on average, they lasted for nearly 100 min on average with target cells coated with 10 μg/ml anti-CD3 Ab. The relationship between contact duration and killing activity was further reinforced by the finding that the CD8⁺ T cells that did not kill target cells at the intermediate concentrations of anti-CD3 Ab (1 and 2.5 μg/ml) were those establishing the shortest contacts (Fig. 5d). Interestingly, the contact duration did not stand as the rate-limiting parameter for killing since most of the contacts associated with a lack of killing were long enough to accommodate delivery of the lethal hit, which we found to be 17 min in average for all three anti-CD3 Ab coating concentrations associated with killing (Fig. 5e). Furthermore, the contact durations between cytotoxic T cells and target cells largely exceeded the time it took for lethal hit delivery, indicating that although adhesion duration appeared to relate to killing efficacy, it did not directly instruct the killing decision process. In order to appreciate how contact dynamics might relate to the efficacy of cytotoxic activity at a population scale, we used automated live imaging to record over 24 h microwells seeded with CellTraceViolet stained CD8⁺ T cells and GFP expressing target cells (Fig. 5f). In agreement with the measurement of contact duration on individual conjugate, our population-wide analysis showed that contact probability increased as a function of anti-CD3 Ab concentration on the target cells (Fig. 5g). Contact frequencies peaked between 2 and 4 h of coculture, which was found to coincide with the time at which the killing of target cells started to be detected (Fig. 5h). Contact frequency appeared to relate to killing efficiency since only the conditions with a clear increase of contact frequency (anti-CD3 Ab concentrations of 0.5, 1 and 5 μg/ml) were associated with efficient killing, defined as a sharp rise in the number of dying target cells in the 3- to 10-h time interval, yielding to >50% target cell elimination after 10 h. The fact that contacts immediately preceded killing and that the frequency of contacts mirrored the efficacy of the ensuing killing activity suggests that the frequency of adhesion, presumably mediated by LFA-1 activation determines the efficacy of the cytotoxic activity. In agreement with the LAMP-1 measurements, global killing activity appeared to be a thresholded response with a sharp rise in killing activity occurring between the 0.1 and 1 μg/ml anti-CD3 Ab concentrations.

## LFA-1 conformational switch is required for lytic granule exocytosis and killing

Our data are strongly suggestive of an involvement of LFA-1 conformation activation in the regulation of CD8⁺ T cell adhesion to target

cells and the decision to degranulate. In order to formally test the requirement of the high-affinity conversion to sustain killing, we specifically blocked the conformation activation of LFA-1, employing the BIRT 377 compound, a LFA-1-specific allosteric antagonist previously shown to block the induction of the high-affinity conformation detected by the m24 antibody[30,31]. As expected, addition of BIRT 377 to CD8⁺ T cells interacting with anti-CD3 Ab-coated P815 target cells efficiently prevented the maturation of LFA-1 to the high-affinity conformation, as detected by m24 Ab staining and flow cytometry analysis (Fig. 6a). The impairment of high-affinity LFA-1 induction was associated with a complete inhibition of lytic granule exocytosis, as measured by LAMP1 surface expression. This inhibition however did not impact the expression of LFA-1 itself, as verified by staining the treated cells with the Hi-111 and TS2/4 Abs (Fig. 6b). Using the automated live imaging setup presented in Fig. 5f, we further observed that BIRT 377 treatment almost completely impaired contacts with P815 cells and cytotoxic activity over a 24-h recording period (Fig. 6c, d). Together these data provide direct evidence that the process of LFA-1 conformation activation is a crucial mechanism that conditions interaction of cytotoxic T cells with target cells, exocytosis of lytic granules and ultimately killing activity.

## ICAM-1 availability conditions adhesion to target cells and killing

After having explored the metrics of LFA-1 activation as a function of TCR stimulation, we asked how ICAM-1 availability might impact TCR-mediated LFA-1 activation and associated killing efficacy. For that purpose, we recorded both low-affinity and high-affinity LFA-1 signals by high-content imaging in cross-titration experiments whereby multiple combinations of ICAM-1 and anti-CD3 Ab coatings were tested. We observed that increasing ICAM-1 densities enhanced the intensity of Hi-111 staining at the adhesion plane independently of TCR engagement (Fig. 7a), suggesting that LFA-1 might align with the availability of ICAM-1 at the immunological synapse via a diffusion process. We noted that TCR stimulation increased closed LFA-1 intensity in the absence of ICAM-1 and at low ICAM-1 densities. LFA-1 conformational activation, as determined with the m24 staining, increased as a function of TCR stimulation strength for all ICAM-1 densities tested but was also directly correlated to the density of available ICAM-1 (Fig. 7b). The calculation of the ratio of the m24 and Hi-111 staining intensities indicates that TCR stimulation promotes comparable conformational activation of LFA-1 once the close LFA-1 densities determined by ICAM-1 are normalized (Supplementary Fig. 6). These data clearly show that the level of LFA-1 activation results from a combined sensing of ICAM-1 density (which appears to set the density of LFA-1 at the contact plane) and strength of TCR stimulation (which determines the proportion of available LFA-1 that may switch to high-affinity conformation). The limited impact of the TCR stimulation on the LFA-1 conformation ratio is in line with the limited proportion of clusters in the open state even upon strong activation. These observations suggest that although the LFA-1 conformation activation operates across a relatively limited range of values it translates into large changes in terms of adhesion. In order to directly assess the impact of ICAM-1 level from the target cell side on CD8⁺ T cell adhesion and killing, murine P815 cells were treated with the murine ICAM-1 blocking Ab YN1/1.7.4[32,33] (Supplementary Fig. 7). In this setting, murine LFA-1 from the P815 cell side is not expected to bind human ICAM-1 from the T-cell side[34]. In a flow cytometry set-up with forced contacts between CD8⁺ T cells and the targets cells, we observed that target cells with blocked ICAM-1 induced about 50% less LFA-1 activation than target cells treated with an isotype Ab (Fig. 7c). The degranulation was also significantly reduced, indicating that lower ICAM-1 availability reduces the capacity of cytotoxic T cells to activate LFA-1 and to secrete lytic granules (Fig. 7d). We then directly investigated the impact of ICAM-1 blockade on the kinetics of adhesion and

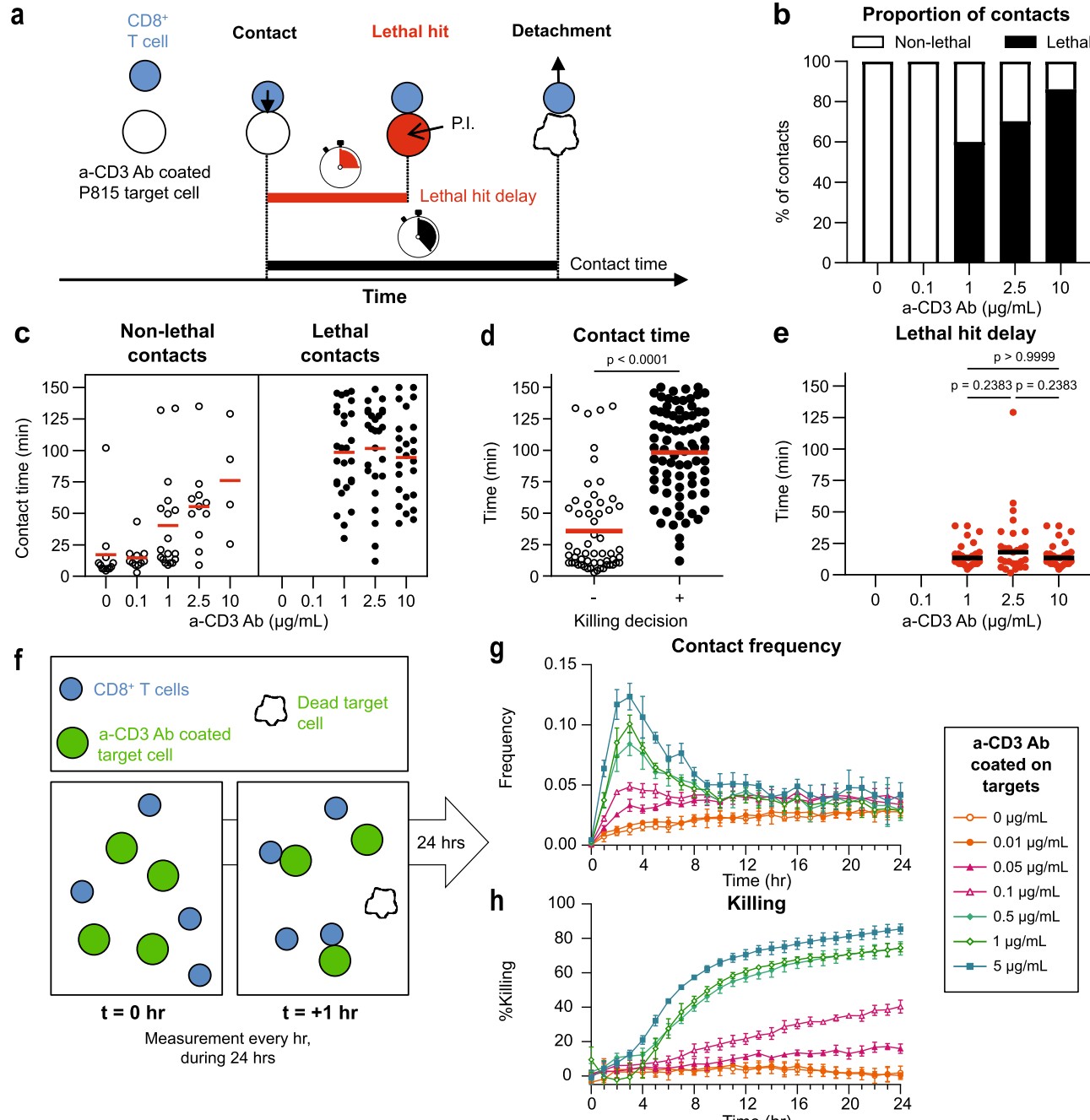

**Fig. 5 | Target cell contact dynamics, lethal hit delay and killing kinetics.**
**a** Graphical representation of the time-lapse microscopy experiments and extracted parameters. P.I. propidium iodide. **b** Percentage of non-lethal and lethal contacts for the indicated anti-CD3 Ab concentrations. Contacts were recovered across three independent time-lapse microscopy experiments (12, 11, 45, 37 and 29 contacts per condition, respectively). **c** Contact time analyzed separately for non-lethal and lethal contacts (same experiments as in **b**). Dots represent individual contacts and bars represent the mean. **d** Contact time analyzed separately for non-lethal ($n = 56$) and lethal ($n = 78$) contacts (same experiments as in **b**). $p$ values calculated by two-sided $t$-tests. **e** Lethal hit delay among lethal contacts. Dots represent individual contacts and bars represent the mean (0, 0, 27, 26 and 25 contacts analyzed). Significance tests are parametric one-way ANOVA. **f** Graphical representation of the high-throughput imaging of contact and killing kinetics. Pre-stained CD8+ T cells and anti-CD3 Ab coated P815 target cells were imaged every hour for 24 h to record contact frequencies and target cell killing. **g** Contact frequency between anti-CD3 Ab coated P815 cells and CD8+ T cells, defined as the number of contacts divided by the number of live P815 cells. Each point represents the mean of four experimental wells, and the error bar is the SEM. The data are from one representative experiment among two independent experiments. **h** Kinetics of target P815 cell killing, as defined by the loss of GFP+ P815 cells compared to control wells containing GFP+ P815 cells alone. Points represent the mean of four replicates and error bars represent the SEM. The data are from one experiment among two independent experiments. Source data are provided as a Source Data file.

cytotoxicity. The contact frequency was significantly decreased for target cells treated with YN1/1.7.4 (Fig. 7e). Killing efficiency was also impacted by ICAM-1 availability (Fig. 7f). The effect of ICAM blockade was further confirmed by monitoring contact frequency and proportion of lethal contact by live microscopy (Supplementary Fig. 8a, b). Together, these data indicate that combined with TCR activation, ICAM-1 availability sets LFA-1 conformational activation, stability of target cell contacts, degranulation and killing.

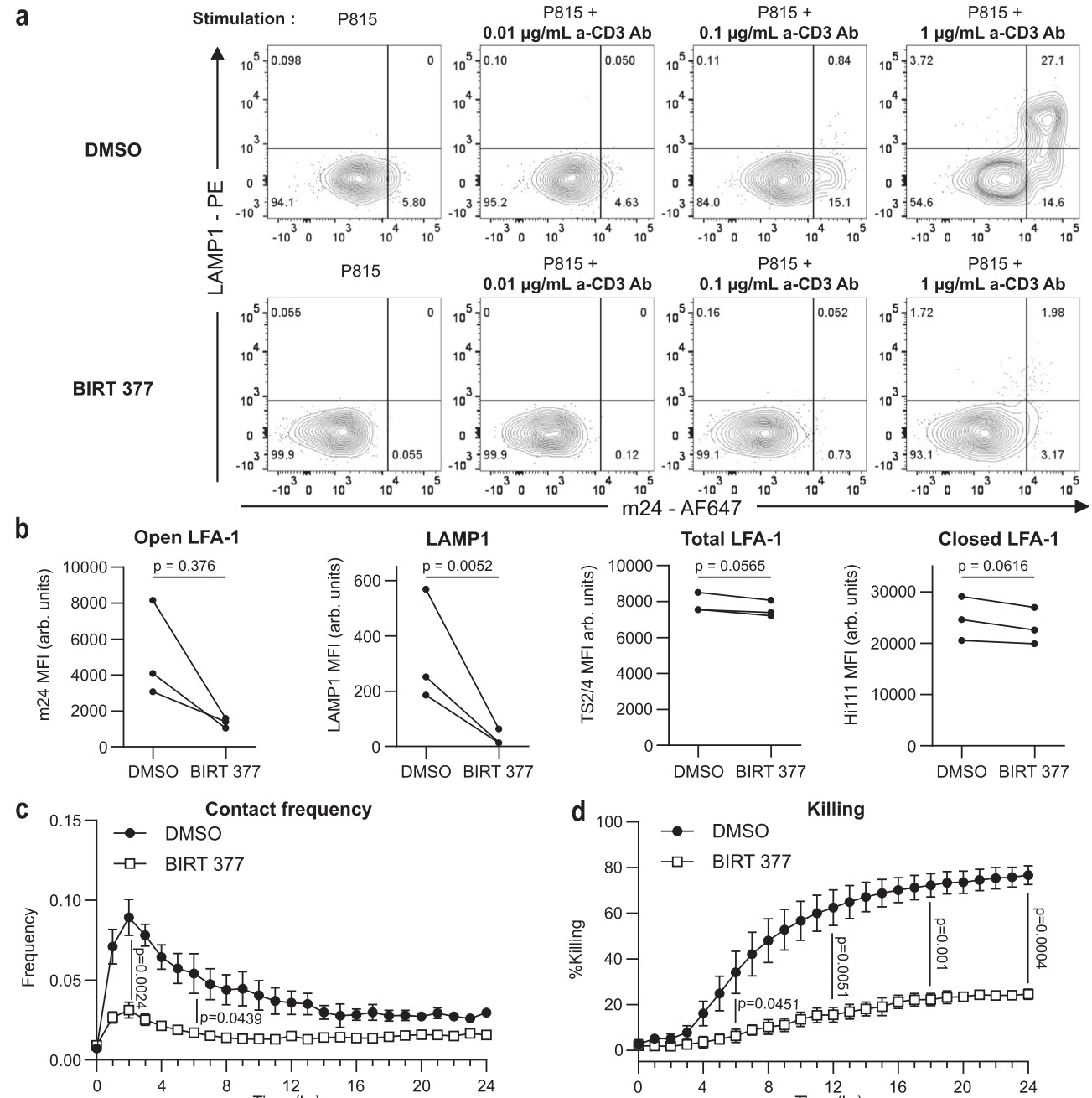

**Fig. 6 | Impact of LFA-1 activation blockade on degranulation, contact frequency and killing. a** Flow cytometry analysis of LAMP1 and m24 expression on CD8+ T cells stimulated by P815 cells coated with the indicated concentrations of anti-CD3 Ab. Top raw displays CD8+ T cells treated with DMSO while bottom raw displays CD8+ T cells treated with 10 µM BIRT 377. One representative experiment among three independent experiments is shown. N = 20,000 cells for each condition. **b** In CD8+ T cells stimulated with P815 cells coated with 1 µg/ml anti-CD3 Ab, MFI of m24, LAMP1, TS2/4 and Hi-111 stainings depending on the treatment (DMSO or BIRT 377) applied to the cells. p values calculated by two-sided ratio paired t-test. Three independent experiments are shown. **c** Contact frequency between P815 cells coated with anti-CD3 Ab (1 µg/ml) and CD8+ T cells depending on the treatment (DMSO or BIRT 377) applied to the cells. The contact frequency is defined as the

number of contacts divided by the number of live targets. Points represent the mean of three independent experiments and the error bars represent the SEM. The significance of the differences between the curves is determined at the 2-h and 6-h time points with two-sided unpaired t-tests. **d** Percentage of killing of the P815 cells coated with anti-CD3 Ab (1 µg/ml) depending on the treatment (DMSO or BIRT 377) applied to the cells. Killing values are calculated from the number of residual alive P815 cells at the considered time points. Points represent the mean of three independent experiments and error bars represent the SEM. The significance of the differences between the curves is determined at the 6, 12, 18 and 24-h time points with two-sided unpaired t-tests. Gating strategy exposed in Supplementary Fig. 5a. Source data are provided as a Source Data file.

## Discussion

The mechanisms governing the activation of the integrin LFA-1 in the context of T-cell stimulation have been extensively investigated. However, how LFA-1 conformational activation scales with TCR stimulation strength and ICAM-1 availability and how it tunes T cell

adhesion and delivery of cytotoxic molecules have remained poorly explored questions. By providing metrics of LFA-1 activation as a function of TCR stimulation and ICAM-1 availability, our study sheds light on the understanding of LFA-1 activation in human cytotoxic T lymphocytes.

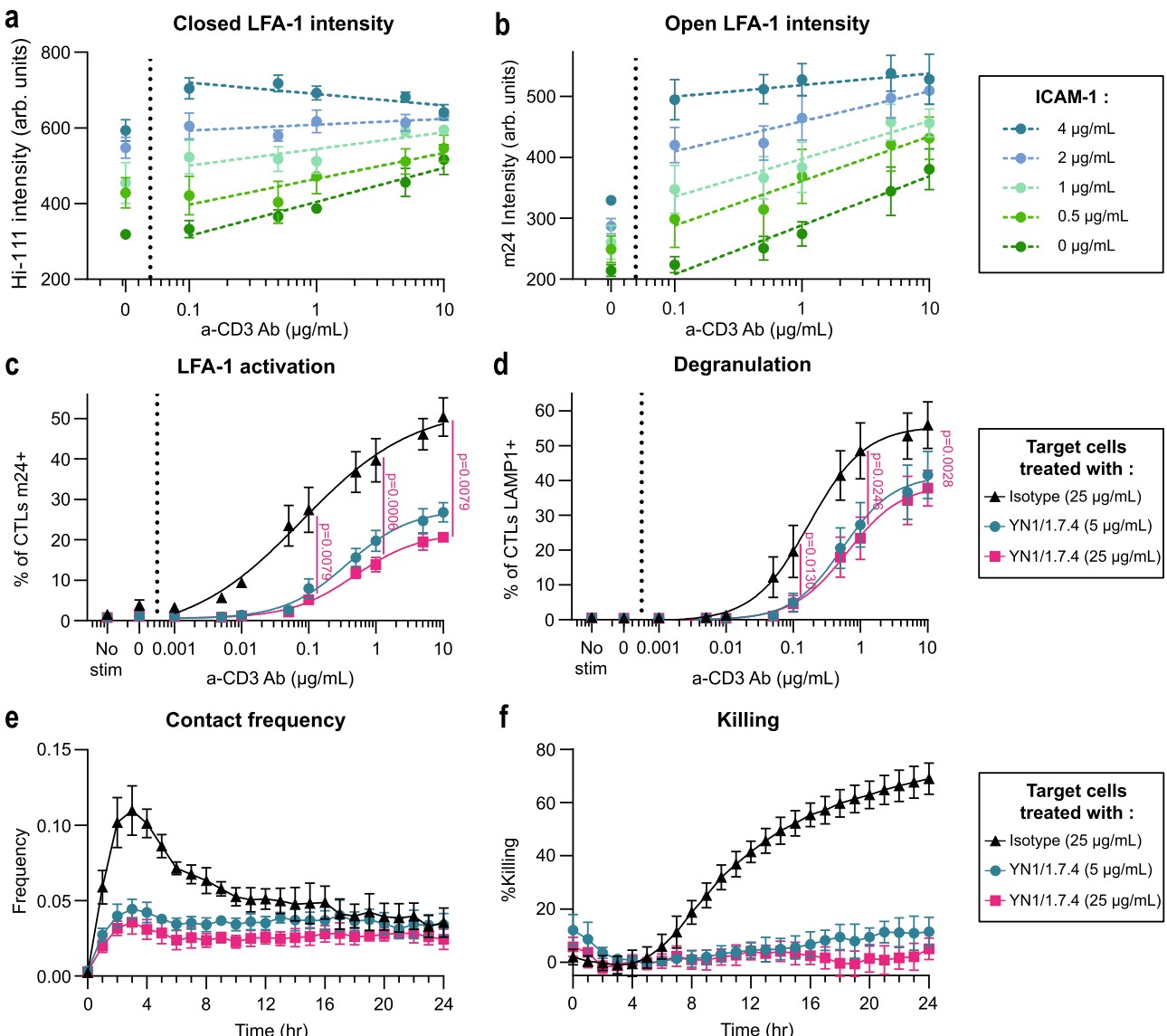

**Fig. 7 | Influence of ICAM-1 availability on TCR-evoked LFA-1 activation.** Fluorescence intensities of the closed (**a**) and open (**b**) LFA-1 conformations, as calculated from high-throughput confocal images of CD8[+] T cells of the same donor spreading on the indicated concentrations of ICAM-1 and anti-CD3 Ab. Each symbol represents the mean and SEM of three independent experiments, each including two replicates. Dotted lines represent linear regression on the mean values type a*log[a-CD3 concentration] + b. Percentage of CD8[+] T cells detected by flow cytometry as positive for surface expression of open conformation LFA-1 (**c**) and LAMP1 (**d**) upon stimulation with P815 cells coated with the indicated anti-CD3 Ab concentrations and the addition of the anti-ICAM-1 Ab YN1/1.7.4. Mean and SEM of

three independent experiments. The lines represent sigmoidal fits. $N = 30,000$ cells per condition. The significance of the differences between the curves is determined with two-sided ratio paired $t$-tests. **e** Contact frequency between P815 cells coated with anti-CD3 Ab (0.2 μg/ml) and CD8[+] T cells, defined as the number of contacts divided by the number of live targets. Each symbol represents the mean and SEM of two independent experiments, each including quadruplicate values. **f** Percentage of killing of the P815 cells coated with 0.2 μg/ml anti-CD3 Ab calculated from the number of residual alive P815 cells at the considered time points. Symbols represent the mean and SEM of two independent experiments, each with four replicates. Source data are provided as a Source Data file.

Through the application of super-resolution microscopy approaches, we reveal that LFA-1 is at least in part confined in discrete clusters that pre-exist in inactivated conformation at the surface of resting effector T cells. This observation challenges the current model of TCR-evoked regulation of T cell activation via an inside-out signal leading to LFA-1 conformational maturation[35]. In line with the current model, the gradual increase in the number of high-affinity nanoclusters may result from the de novo clustering of isolated molecules concomitantly to the conformational maturation process. This scenario could be facilitated by the apparent reserve of non-clustered closed conformation molecules or the mobilization of intracellular pools[36]. The fact that we hardly observed mixed clusters (using 3 super-resolution imaging modalities) suggests that if de novo clustering is the main source of

open conformation LFA-1 nanoclusters, it is based on a very efficient cooperative mechanism. On the other hand, our finding that closed and open conformation nanoclusters have a comparable size argues against the initial notion that TCR may regulate LFA-1 mediated adhesion via a process of avidity modulation[7,37]. Our data are suggestive of nanoclusters acting as switches that change conformation in a digital fashion upon TCR engagement. The fact that the densities of detected closed and open nanoclusters scaled proportionally to the global staining intensities across the stimulation conditions further argues against an active clustering process resulting from the TCR engagement. The alternative cluster switch model is however in apparent contradiction with the fact that the number of closed conformation nanoclusters remained stable with increasing anti-CD3 Ab

concentrations, unless one considers that the conformation switch of individual nanoclusters is compensated by the enrichment of closed conformation nanoclusters at the synapse plane. This is possible given our imaging flow cytometry data showing that while open-conformation LFA-1 is mostly confined within the synapse, closed-conformation LFA-1 remains at equilibrium between the synapse and the rest of the cell. This suggests that a diffusion-trapping process allows replenishing the synapse with LFA-1 to compensate for its engagement and activation. Such process is actually compatible with both the de novo cluster assembly model and the cluster switch model. The activation of LFA-1 within nanoclusters might be further regulated by the possibility that LFA-1 engages with ICAM-1 in cis as recently described in neutrophils[38]. If at work in T cells, such mechanism could keep nanoclusters on hold and contribute to the scalability of LFA-1 engagement with target cell ICAM-1. Beyond our study, it would be interesting to further study the chemical/physical mechanisms underlying the assembly and/or switch behavior of integrin nanoclusters and explore if it applies to other receptors[39].

Our further data suggest that LFA-1 nanoclusters operate as detection units to sense both ICAM-1 and TCR ligand densities. Indeed, ICAM-1 density appears to set the concentration of LFA-1 molecules available at the synapse, while TCR triggers a tunable conformation activation of available LFA-1. Our findings are in line with previous observations that the binding of T cells to antigen-presenting cells grows with both the strength of signals delivered by p-MHC complexes and the density of ICAM-1[18]. The fine-tuning of LFA-1 activation sustained by >1000 nanoclusters is probably able to integrate additional cues including those arising from receptors able to activate inside-out signaling such as chemokine receptors[40] or receptors shown to associate to LFA-1 such as DNAM-1[41]. This might also apply to the integration of magnesium concentration recently shown to set the magnitude of LFA-1 conformational activation in cytotoxic T cells and thereby to control target cell killing[42]. Although not assessed in this work, we propose that LFA-1 nanoclusters may also operate as force-transmission units and that individual cluster size is probably calibrated for optimal force generation. Indeed, many small clusters, rather than large adhesions, are expected to provide a flexible system to finely tune the stop and go behavior of lymphocytes[43], to adapt to the irregularities of target cell surfaces[44] and to generate local forces for piecemeal control of granule exocytosis[14].

From a physical point of view, organization in dense, multi-LFA-1 functional units presents at least two advantages. Firstly, pioneering work by Bell indicated that coordinated ligand-receptor binding of many receptors is more efficient than just the sum of the individual efficiencies of single ligand-receptor couples[45,46]. To this respect, other adhesion molecules such as cadherins have been observed to dwell in small clusters to perform their adhesion function[47]. Secondly, the possibility that clusters may operate via an all-or-nothing conformation switch is reminiscent of a cooperative behavior in integrin binding, generically resulting from the too high-energy cost of having close-neighbor molecules with different conformations. In the present study, we show that closed, unbound integrins coexist on the T cell membrane with open, bound ones. Cooperativity of integrins when binding to their ligands has been observed on fibroblasts, by controlling the inter-ligand distance through nanopatterning[48]. Cell adhesion was strongly enhanced when the distance between ligands went below ~70 nm, indicating that not-too-distant integrins, as in clusters, tend to bind cooperatively to their ligands. Cooperativity has been shown to result (at least in part) from the elastic cell membrane and glycocalyx cooperative deformation[49] and thermal membrane roughness reduction upon binding[50]. As a corollary, availability of the ligand on the apposed membrane enhances cooperativity by stabilizing the open conformation, and this is consistent with our observation that lower availability of ICAM-1 reduces LFA-1 cluster activation.

We have also observed that the enrichment in open conformation nanoclusters occurs gradually along a wide range of TCR stimulation strengths. This may indicate that TCR-evoked LFA-1 activation is a short-range processes, as expected from a mechanism propagated by the actin cytoskeleton. This implies that if only few TCR clusters[51] are stimulated, they can activate only few LFA-1 clusters in their neighborhood. Indeed, very recent experiments using localized, micrometer-sized OKT3 and ICAM-1 spots[14] accredit the hypothesis that both LFA-1 recruitment upon TCR engagement, and degranulation in presence of both ligands, are short-range mechanisms, at a micrometric scale substantially smaller than the synapse itself. It must also be emphasized that only a finite fraction of LFA-1 fluorescence intensity was detected to belong to clusters, as identified from the STED images. The remaining LFA-1 molecules belong to a background that could correspond to a dilute phase generically coexisting with the more condensed cluster phase, as predicted by theoretical works[52]. However, the STED resolution is insufficient to identify single LFA-1 molecules and one cannot certify the exact nature of the LFA-1 background.

Interestingly, our data indicate that change of morphology from a motility-associated shape with protrusions to a more roundish shape precedes adhesion in terms of TCR stimulation and that adhesiveness to the target cells is required but not sufficient to allow lytic granule exocytosis and killing. Our study reveals distinct TCR stimulation thresholds required (1) to halt and likely scan a target of interest, (2) to trigger LFA-1 conformation activation and (3) to promote degranulation. The response functions between these functional states are rather abrupt, being well-shaped sigmoids in the two latter cases. These observations are notably supportive of the notion that cells ensure high-quality contacts with target cells before they activate the degranulation process. Furthermore, live microscopy measurements showed that the mean non-lethal contacts time between target cells and T cells is longer than the duration needed for the lethal hit delay, suggesting that when the adhesion is promoted, the time is not the limiting factor, in line with the above observations.

The present study was conducted on ex vivo expanded effector CD8+ T cells purified from healthy individuals. Future application of the herein developed analytical pipeline to primary cells isolated from tumors might reveal alterations in the TCR-LFA-1 cooperative mechanism[53]. Additionally, the TCR-LFA-1 cooperation could be investigated in the context of CAR T cells, in which the chimeric receptor might not signal as efficiently as the native TCR and might explain sub-optimal synapse assembly[54,55].

## Methods
### Cell lines
Untransformed CD8+ T cell lines were generated from blood samples of healthy donors, which were collected following standard ethical procedures (Helsinki protocol) and as per French Bioethics law. This included obtaining an informed consent from each of the donor. The work on blood samples conducted in this study was covered by an approval by the local ethics committee (Comité de Protection des Personnes Sud-Ouest II et Outre-Mer II). PBMC were isolated by density gradient centrifugation over a Ficoll gradient (PAA). CD8+ T cells were isolated by negative selection (Stem Cell) according to the manufacturer's instructions, reaching a purity >95%. CD8+ T cells were cultured in RPMI 1640 glutamax (Gibco) supplemented with 5% human serum (PAA), 100 μg/ml penicillin/streptomycin, 10 mM Hepes, 0.1 mM MEM nonessential amino acids and 1 mM sodium pyruvate (all from Invitrogen). Cells were stimulated every 2 weeks in the presence of $1 \times 10^6$/ml irradiated allogeneic PBMC and $1 \times 10^5$/ml irradiated EBV-transformed JY cells with 1 μg/ml PHA, 100 IU/ml rhIL-2 and 5 ng/ml rhIL-15. Blood samples were obtained following standard ethical procedures (Helsinki protocol). P815 target cells were maintained in vitro with DMEM (Gibco) supplemented with 10% FCS (PAA), 100 μg/ml

penicillin/streptomycin, 10 mM Hepes, 0.1 mM MEM nonessential amino acids and 1 mM sodium pyruvate (all from Invitrogen). P815 cells were transduced with a GFP encoding lentiviral vector and FACS sorted for GFP expression.

### STED microscopy

Circular microscope cover glasses (Marienfield superior) were coated overnight with ICAM-1 (R&D systems) at 2 µg/ml and anti-CD3 Ab (OKT3 clone, Invitrogen) at concentrations between 0 and 10 µg/ml. CD8[+] T cells were seeded on the cover glasses at 37 °C and stained 5 min later with conformation-specific anti-LFA-1 Ab: m24 (coupled to biotin, Biolegend) and Hi-111 (coupled to AlexaFluor594, Biolegend), each used at 10 µg/ml during 15 min. Cells were fixed with 4% paraformaldehyde for 15 min at 37 °C. Streptavidin STAR RED (Abberior) was used at 9 µg/ml to reveal the m24-biotin Ab. Nuclei were stained with DAPI at 10 µg/ml for 5 min. In control experiments, cells were stained with 100 nM MemBright 590 (idylle labs) post fixation for 10 min at room temperature to reveal plasma membrane lipids. The images were acquired with a STED microscope (Leica SP8 equipped with STED 3X), using a 100X/1.4 oil objective and the LSX software. The wavelengths of the laser used were 532 and 635 nm for the excitation, and 775 nm for the extinction. The images were pre-treated on Fiji with a background subtraction and a median filter in order to reduce the noise. In a set of control experiments aiming at assessing LFA-1 organization at a fully resting cellular state, CD8[+] T cells were washed with PBS and kept in suspension by gentle mixing while being fixed with 4% paraformaldehyde (15 min at 37 °C). Following washing, cells were stained with Hi-111 Ab coupled to AlexaFluor594 (Biolegend), at 10 µg/ml during 15 min. Following washing, cells were projected on circular microscope cover glasses (Marienfield superior) by centrifugation ($422 \times g$ for 5 min). STED images were acquired similarly to stimulated cells, as described above.

### LFA-1 nanocluster segmentation algorithm

The trainable Weka segmentation plug-in of Fiji[56] was trained in order to create a classifier recognizing activated nanoclusters on the m24 Ab channel and a classifier recognizing inactivated nanoclusters on the Hi-111 Ab channel. Each classifier was trained on STED pictures of 14 CD8[+] T cells across conditions. The classifiers were then applied to STED pictures of CD8[+] T cells (13-14 cells per condition) in order to detect nanoclusters and their properties (position, boundaries, area, composition). In order to test the accuracy of the algorithm at detecting nanoclusters, it was applied to STED pictures of CD8[+] T cells stained with the MemBright membrane probe, which inserts in the lipid bilayer via lipid anchors[57]. A step of erosion was first applied to remove the cell edges presenting with enriched membrane folds and accumulation of the probe. In the resulting synaptic areas, although the algorithm detected local density fluctuations as clusters, their density was very reduced as compared to that of LFA-1 nanoclusters (Supplementary Fig. 9a–c). In agreement with a previous STED analysis, such events might correspond to local variations of lipid distribution at the plasma membrane such as lipid rafts[58]. A complementary approach to test the accuracy of the algorithm consisted in generating an artificial image containing about 1000 nanoclusters presenting the characteristics of LFA-1 nanoclusters, as well as a background of randomly distributed individual molecules (Supplementary Fig. 9c–f). In accordance with the MemBright staining, the algorithm trained on STED images of LFA-1 stained T cells, detected less than 10% of false positive events in addition to the nanoclusters, due to statistical density fluctuations (Supplementary notes).

### High-content cell imaging

Imaging plates (384-well plates CellCarrierUltra, Perkin Elmer) were pre-coated overnight with crossed concentrations of ICAM-1 (0; 0.5; 1; 2; 4 µg/ml, R&D systems) and anti-CD3 Ab (0: 0.1; 0.5; 1; 5; 10 µg/ml,

OKT3 clone, Invitrogen). CD8[+] T cells were seeded in the plate at 37 °C and stained 5 min later with conformation-specific anti-LFA-1 Ab: m24 (coupled to AlexaFluor647, Biolegend) and Hi-111 (coupled to AlexaFluor488, Biolegend), each used at 1 µg/ml during 15 min. Cells were fixed with 4% paraformaldehyde for 15 min at 37 °C and stained overnight by AlexaFluor546-coupled phalloïdin (Thermo Fisher) at 0.4 µg/ml. Nuclei were stained with DAPI at 10 µg/ml for 5 min. Pictures were acquired with an automated high-content confocal microscope (Opera Phenix, Perkin Elmer) and analyzed by the Harmony software (Perkin Elmer) as described earlier[59].

### Assessment of degranulation by cytometry

GFP-expressing P815 target cells were coated with the indicated concentrations of anti-CD3 Ab (OKT3 clone, eBioscience) during 1 h at 37 °C/5% $CO_2$. The target cells were washed twice in PBS after coating. They were co-cultured with CD8[+] T cells in 96-wells flat plate for 2 h at 37 °C/5% $CO_2$ (30,000 P815 cells and 60,000 CD8[+] T cells). Anti-LAMP1 Ab (PE, BD Biosciences) has been added from the outset of the co-culture, and m24 Ab (Af647, Biolegend) and viability dye (APC-H7, eBioscience) has been added to the wells 20 min before the end of the experiment. The CD8[+] T cells were stained with an anti-CD8 Ab (PerCP Cy5.5, Sysmex) for 20 min. Samples were acquired on Fortessa (BD Biosciences) and analyzed with FlowJo software (FlowJo, Ashland, Ore). The cells were gated in order to study alive (viability dye negative cells) and CD8[+] T cells only.

### Short-term cytotoxic activity

P815 target cells were coated with the indicated concentrations of anti-CD3 Ab (OKT3 clone, eBioscience) during 1 h at 37 °C/5% $CO_2$. The target cells were washed twice in PBS after coating. CD8[+] T cells were stained with CMFDA during 30 min. P815 target cells and CD8[+] T cells were transferred at a 2:1 ratio in the wells of a 8-wells plate IBIDI pre-coated with 10 µg/ml fibronectin. Propidium iodide was added at 100 nM in order to visualize the apoptosis. Well content was recoded every minute by a spinning disk microscope equipped with temperature and $CO_2$ control (37 °C/5% $CO_2$). The formation of conjugates was studied with the 488 and brightfield channels in superposition, to detect CD8[+] T cells and P815 target cells, respectively. Each time a CD8[+] T cell contacted a P815 cell, an image crop was done to isolate the event and the contact time was determined manually with the time-points identified on the stacks.

### Long-term cytotoxic activity

GFP-expressing P815 target cells were coated with the indicated concentrations of anti-CD3 Ab (OKT3 clone, eBioscience) during 1 h at 37 °C/5% $CO_2$. The target cells were washed twice in PBS after coating. CD8[+] T cells were stained with 2.5 µM CellTraceViolet (Thermo Fisher) during 10 min. P815 target cells and CD8[+] T cells were transferred (2500 P815 cells and 5000 CD8[+] T cells) in the wells of a 384-well plate (Perkin Elmer) precoated with 1 µg/ml fibronectin. Aphidicolin (Sigma) was added at 0.1 µg/ml to prevent target cell proliferation and allow net measurement of CD8[+] T cell-mediated cytotoxicity. Well content was recoded every hour for 24 h with an automated high-content confocal microscope (Opera Phenix, Perkin Elmer) set to 37 °C and 5% $CO_2$. Cytotoxicity was analyzed by counting the number of alive target cells as detailed previously[60].

### Blockade of LFA-1 conformational activation

The CD8[+] T cells were pre-treated with 10 µM BIRT 377 (R&D systems) or DMSO for 30 min. The P815 target cells used for the degranulation and killing experiments were re-suspended in the corresponding media (containing either 10 µM of BIRT 377 or DMSO) immediately prior to the co-culture with the pretreated T cells, in order to maintain the treatment during the experiments.

## Statistical analysis

Statistical analyses were performed using GraphPad software. Statistical significance was determined by performing one-way ANOVA tests or *t*-tests (the statistical analyses used throughout the manuscript have been indicated in the figure legends). Resulting *p* values are indicated directly on the figures.

## Reporting summary

Further information on research design is available in the Nature Portfolio Reporting Summary linked to this article.

## Data availability

Source data are provided with this paper, as indicated in the figure legends. They can accessed with the following link: https://doi.org/10.6084/m9.figshare.24299671. Additional image data that support the findings of this study are available from the corresponding authors upon request. Data are located in controlled access data storage at the INSERM Infinity Institute and the Theoretical Physics Laboratory of the University of Toulouse.

## Code availability

The code used to generate artificial STED images is provided with this paper. It can be accessed with the following link: https://doi.org/10.5281/zenodo.10084714.

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

## Acknowledgements

We wish to thank Simon Lachambre and Lhorane Lobjois from the INFINITy microscopy platform, as well as Fatima-Ezzahra L'Faqihi-Olive and Anne-Laure Iscache from the INFINITy cytometry platform. We are grateful to Johannes Huppa for technical advices and discussion. This work was supported by FWF (project P 34118-B to L.D.) and CNRS (IRP project SysTact to L.D.). The funders had no role in study design, data collection and analysis, decision to publish, or preparation of the manuscript.

## Author contributions

C.L. performed the experiments, analyzed the results and wrote the manuscript; B.C. contributed high-content cell imaging experiments; D.G. contributed live-microscopy experiments; R.H. contributed super-resolution microscopy experiments; V.D.-E. contributed flow imaging experiments; S.A. provided expertise in STED microscopy and image analysis; N.D. designed the research, supervised the analytical work and wrote the manuscript; L.D. designed the research, supervised the experimental and analytical work and wrote the manuscript.

## Competing interests

The authors declare no competing interests.
