## [Peer Review File · Nature Communications]

LFA-1 nanoclusters integrate TCR stimulation strength to tune T-cell cytotoxic activityREVIEWER COMMENTS

Reviewer #1 (Remarks to the Author):

In this manuscript, Lacouture et al. use super resolution microscopy, together with conventional confocal and time-lapse microscopy, as well as flow cytometry to show that: (1) open and closed conformation LFA-1 appears in pre-formed nanoclusters on the surface of CD8 T cells; (2) T cell activation switches LFA-1 from closed to open conformation; (3) open conformation LFA-1 correlates with CD8 degranulation and killing and (4) that LFA-1 – ICAM1 interaction is required for efficient killing.

While some findings confirm earlier work (e.g. the dependency of efficient killing on LFA1-ICAM1, the role of contact duration) the manuscript contains important findings about the nanoscale organisation of LFA-1 and the kinetic relationship between TCR signalling, integrin activation and degranulation.

Main points:

1.) related to Figure 2: the main point of the authors here is that there appears to be a discrete 'pool' of LFA-1 nanoclusters that switches from closed to open conformation upon TCR stimulation. However, in line 136 (referring to Figure 2B) it is stated that the number of closed conformation LFA-1 clusters stays the same. Isn't this contradictory? If there is a finite pool the number of closed conformation LFA-1 clusters should go down while the number of open conformation should go up. Nevertheless, Figure 2C shows that the fraction of open conformation LFA-1 is going up. Doesn't this suggest that there is considerable de novo assembly of open conformation nanoclusters?

Also related to Figure 2 and Figure S2: although the mean of the cellular area is staying constant, given the considerable spread of the data and the relatively low sample size (which is understandable given the used imaging technique), I would like to see these data normalized to the area of the cells (normalized number of clusters). Does this give cleaner data with a smaller standard deviation? This might also enable the authors to pool the data from their two independent experiments of which only one is shown.

2.) related to Figure 7: one experiment that might be interesting to do here is to vary the levels of ICAM1 on the target cells. The initial, relatively small differences despite full blockage of ICAM1 in Figure 7C & D might indicate that the system is quite robust against changes in ICAM1 levels.

3.) P815 is a mouse cell line and thus expresses mouse ICAM-1 bound by the YN1/1 antibody (PMID: 2981916). This is fine as mouse ICAM-1 binds well to human LFA-1, but mouse LFA-1 expressed on P815 will bind poorly to human ICAM-1 (PMID: 2199576). P815 also lacks CD58, the other major adhesion ligand involved in CTL mediated killing with most human targets. Thus, this is actually a well-chosen target to isolate CTL LFA-1 to target ICAM-1 role in killing. These issues should be presented in explaining why P815 was chosen or discussed in relation to the point below; YN1/1 antibody will bind only to the P815 cells and not to ICAM1 on the human CTL. This is important because someone choosing a human target would probably get different results, even using antibodies to human ICAM-1.

4.) mAb 24 is a ligand induced binding site on LFA-1 beta subunit (PMID: 7685913). So the basal mAb 24 signal when there is no ICAM-1 on the substrate, or ICAM-1 on the target is blocked, is due to cis interaction with ICAM-1, which take place within LFA-1/ICAM-1 nanoclusters as documented on neutrophils (PMID: 30605669). Activated T cells express ICAM-1 that could engage LFA-1 in cis. It is possible that cis ICAM-1 interactions may participate in the mAb 24 binding within the activated nanoclusters. It would be helpful to cite the Ley lab paper and at least discuss its implications if not investigating whether the activated LFA-1 clusters contain ICAM-1 or not. This should not be required, but I would be very interested to know the result!

5) Are you certain that mAb 24 + Hi-111 would be equal to all LFA-1, for example, as seen by TS2/4? Hi-111 has a 10-fold higher affinity for closed than open I domain, but mAb 24 is seeing a confirmation that is physiologically generated by ligand in the beta I like domain, so there may be open I domain where the mAb 24 epitope is not generated as there was no ICAM-1 bound at the point of fixation. You are doing nice work with flow cytometry- can some book keeping be done this this approach to determine how many mAb 24 sites, Hi-111 and TS2/4 (or some other pan LFA-1, non-blocking, epitope). This might help understand how both Hi-111 and mAb 24 sites can increase in some conditions. That may be related to recruitment of LFA-1 from other parts of the cell or due to some "stealth" LFA-1 that can give rise to both Hi-111 and mAb 24 positive under the correct conditions. Have the authors looked at the utility of AL-57, which binds active conformation of the I domain in a Mg²⁺ dependent manner.

Reviewer #2 (Remarks to the Author):

Lacoutre et al. have studied the organization of the "closed", i.e., low affinity and open, "high" affinity states of LFA-1 molecules, using STED microscopy and higher-throughput confocal microscopy-based imaging. The formation of LFA-1 nanoclusters has previously been reported in the context of T-cell migration (see Shannon et al., JCS, 2020), but the relationship between LFA-1 nanoclusters and conformational changes has not been addressed, especially in the context of T-cell synapse formation. Lacoutre et al. propose that LFA-1 forms distinct nanoclusters containing only one or the other states, that the number of open-state nanoclusters scales with the level of TCR signaling and, importantly, that this involves a concerted "switch-like" mechanism wherein a given closed-state nanocluster converts en bloc to an open state. They go on to suggest that this switch conditions T-cell/target contact stability, lytic granule exocytosis, and killing. The experiments are thought-provoking and, for the most part, well performed and the paper is well written. Nevertheless, I have important reservations concerning the methods used and the interpretation of the data, that undermines my confidence in the significance of the findings. Overall, the authors' claim that a switch-like opening of LFA-1 nanoclusters licenses killing of target cells is far from being convincingly demonstrated.

First, the imaging experiments were done on glass, which is problematic insofar as glass has been shown to be capable of activating T cells, and poly L-lysine even more so (Santos et al. NI, 2018). So, the issues then are: what does the true resting state look like, and is it safe to conclude that LFA-1 is pre-clustered in the resting state at all? It could well be that only very small numbers of closed-state nanoclusters, if any, exist in the truly resting state, and so the notion that activation is associated with the conversion of existing nanoclusters would be incorrect. The authors refer to work by others claiming that LFA-1 is pre-clustered, for example Baumgart et al., but Baumgart et al. also worked from glass surfaces. To establish the starting organization, it would be necessary to keep the cells free from the influence of all surfaces. At the very least the distribution in the upper surface of a cell would be worth determining.

Second, notwithstanding the above, I wasn't wholly convinced that the STED data actually supports the notion of clustered closed-state LFA-1 in Fig. 1A. Certainly, the segmentation algorithm produces nanoclusters, but there seems to me to be a step difference in the degree of local intensity variation for the closed- and open-state staining, and in the area of the cell surface across which the intensity varies. Without controls, it's not clear that the

closed-state nanoclusters picked by the algorithm aren't artefactual. What's needed here would be, e.g., a membrane stain, or some other protein not thought to be clustered. The Baumgart et al. paper could be used to identify suitable controls and it would be helpful to the authors if the algorithm failed to cluster these. It would also have been helpful to have performed calibration experiments to relate signal intensity to protein number, e.g., by performing STED on low densities of the antibodies adsorbed to glass, so that it can be certain that clusters are really forming. It's also unclear from the methods whether the diffuse non-clustered LFA-1 signal truly reflects monomeric protein or non-specific binding of the antibody, as the authors do not indicate whether the cells were washed prior to fixation/imaging.

Third, and perhaps most importantly, I was unconvinced by the central claim of the paper, which is that there's "digital" conversion of closed-state nanoclusters to open-state ones. All things being equal, if there was digital conversion of closed-state nanoclusters to open-state ones, the number of closed-state nanoclusters would decrease in proportion to the increase in open-state nanoclusters, but this is not what is seen: both increase, albeit slowly in the case of the closed-state LFA-1. I think that, considering also the data in Fig. S5, it's more likely that there is an increase in open-state nanoclusters, formed from the pool of unclustered closed-state LFA-1. This would fit with the conventional view of cluster formation.

Fourth, in my view the authors fail to establish a strong link between adhesion or killing and the conversion of nanoclusters from closed to open, if it is assumed that this occurs. It must be said that the correlation between the gradual changes observed using STED (Fig. 2D) marking nanocluster "conversion", and the sigmoidal responses observed, e.g., for degranulation (Fig. 4B) and, especially, doublet formation (Fig. S4B), with anti-CD3 concentration is poor. The correlations between the overall levels of each LFA-1 state measured confocally and, e.g., degranulation versus anti-CD3 concentration is far more convincing. The problem, of course, is that it's impossible to know which form of LFA-1 is producing the key effects in the functional assays (assuming the measurements on glass are meaningful). Is it the relatively small numbers (20%) of LFA-1 proteins in nanoclusters or the large number of unclustered molecules. These issues are compounded by the fact that attempts weren't made to determine whether the cells that were imaged using STED had actually triggered, and by only a single time-point (20 min) being examined, leaving uncertainty about the true dynamics of nanocluster "conversion" and its real dependence on TCR signaling, on a per cell basis.

Finally, I was struck by the very small effect of the anti-CD3 antibody on the ratio of closed/open state LFA-1 measured confocally in Fig. S5, in the absence of ICAM-1 (but also in the presence). This looks like an ~10% shift. What does this say about inside-out signaling? Doesn't this undermine the thesis that the TCR oversees LFA-1?

Minor points/questions

1. Line 56: what's meant by "metrics of TCR/LFA-1 cooperation"?
2. Line 80: what's meant by "discretization of signals into nanoswitches"? What is "discretization".
3. Line 94: "pictures" should be "images".
4. Line 96: if there's only three-fold enrichment of LFA-1 in the 80-90 nm diameter clustered areas, what is it that qualifies as a nanocluster (also see comments above about the stoichiometry of the clusters)? How would a random distribution look according to the segmentation algorithm? Would a simulation be helpful?

5. Line 101: what's driving the formation of closed nanoclusters (if they form)?
6. Line 168: rather than "cells with higher surface LFA-1 expression are more prone to activate it" it's clear that the ratio of closed/open states is constant for all levels of LFA-1 expression in Fig. 3C, suggesting that the cells are equally prone to activate the same fraction of LFA-1, irrespective of expression level; it's just that there's more of it for the higher-expressing cells.
7. Line 328: the use of "smoothly" is inappropriate.
8. Line 441: the HCS acronym should be defined.
9. Line 441-442: what temperature and CO₂ level was used?
10. Fig. 1: have the (bivalent) antibodies altered the clustering of LFA-1? Would the results be similar if the LFA-1 was labelled with a genetically-encoded tag or using Fab fragments?
11. Fig. 2: it would be helpful to know the contribution of inside-out signaling vs. ICAM-1 engagement on driving the arrangements observed, by comparing open conformation nanoclustering on glass with and without ICAM-1.
12. Figs 4, 5: a control with no LFA-1 engagement/blocking LFA-1 ought to be included in these experiments. Elsewhere it has been shown that ICAM-1 is dispensable for T-cell mediated killing of tumour cells (see Regev et al. *Frontiers Immunol*, 2022).
13. Fig. 4A: I think the second row should start with a concentration label of 0.1 µg/ml.
14. Fig. 4c: how was the threshold set for analysing LAMP+ cells for the MFI measurements shown? This should be indicated as a supplementary figure or in the methods.
15. Fig. S4D: what are we to make of the 60% of LAMP+ cells that are neither m24+ or m24-? Surely, the fraction of cells that are either positive or negative for any given label is 100%.
16. From Fig. S5 it appears that the presence of ICAM-1 has only very modest effects on the ratio of open to closed LFA-1. Is ICAM-1 being engaged properly on the surface? When adsorbed to glass it will likely adopt multiple positions, some more/less accessible than others to LFA-1. This possibility should be acknowledged.
17. More experimental detail should have been included in the Methods. Were the cells ever washed after staining with OKT3 (lines 416/417, 426/427)? OKT3 in solution can block further OKT3 binding and, potentially, induce signaling. Simply stating that a 2:1 ratio of cells was used (line 428 and 438) is insufficient, when cell density could impact the dynamic range of cell conjugation/killing experiments; actual numbers and densities should be noted. What was the concentration of CellTrace violet used (line 437)? Also, it's indicated that contact time, contact number and lethal cytotoxicity were manually quantified (line 431-432). But how were the cell contacts detected/defined? Was this using brightfield? More details are needed to appropriately judge whether manual analysis was a reasonable approach or not.

Reviewer #3 (Remarks to the Author):

It has been long established that T cell clustering represents a mechanism by which T cell differentiation and effector function is controlled. Some studies have shown that LFA-1 undergoes conformational changes upon binding to ICAM-1, leading to the formation of stable adhesion complexes between T cells and target cells. The strength of the adhesion can affect the sensitivity of T cells to antigen stimulation, as well as the duration and intensity of T cell activation. In addition, the density of ICAM-1 on target cells can influence the degree of LFA-1 clustering and signalling, which can impact T cell activation and killing.

One study published in the *Journal of Immunology* in 2015 (PMID: 26324742) found that the

strength of T cell receptor signalling was positively correlated with the density of ICAM-1 on target cells and that LFA-1 played a critical role in this process. Another study published in *Immunity* in 2019 (PMID: 30893595) showed that LFA-1 signalling could be modulated by the strength of T cell receptor signalling, and that this modulation was important for regulating T cell activation and killing.

In this manuscript, the Lacouture et al, provide evidence that LFA-1 conformation activation scales with TCR stimulation strength to calibrate killing of target cells, and that LFA-1 'licenses' rather than controls the killing decision. The results are very interesting and will be of interest to the field.

Overall, it is an interesting study, and used lovely microscopy to tease apart the role of LFA-1 in T cell function. However, I do have some questions as to the robustness of the data (ie. Only analyzing 14 cells) and the interpretation, given the models used in this system were non-physiological stimulation (OKT3), rather than TCR-mediated, and controlling for affinity and role of other molecules such as CD8.

Here are some of the issues that require addressing in the manuscript:

- ICAM availability on the target cells is a crucial component. Can the authors also justify the choice of using ICAM-1 conc.– how does this concentration compare the physiological levels – Is it saturating?
- The authors perform a lovely STED analysis to show that open LFA-1 nanoclusters scale with TCR stimulation strength. In Figure 2 and S2 – It is not clear in Figure 2B whether the zero CD3 concentration, have also need stimulated with ICAM alone, as per Figure S2, or whether this group is just T cells on glass with no PLL, ICAM, or CD3. It is very hard to follow and requires further labelling, or detail in the figure legends. The methods seem to indicate that the zero concentration of CD3, also has ICAM present at 2 ug/mL and if this is the case, this should be stated in the legend. Where there statistical tests applied to the difference in cluster number between the PLL alone, and ICAM as they look statistically different. And if so, would the authors expect this? These types of analyses should really be performed on several human donors, and on more than 14 cells, to enable robust statistical power to be obtained. Did the authors conduct a power calculation and can justify why they believe only 14 cells was sufficient to draw conclusions?
- Have the authors considered blocking the conformational change of LFA-1 using either peptide inhibitors, monoclonal antibodies, or small molecule inhibitors? This could be a critical element for this study to improve impact and definitely show that it is the LFA-1 change that is tuning the TCR.
- There is not sufficient detail in figure legends to determine exactly what data points are on the graphs and needs some careful attention. For example, Figure 3B authors state, "Mean and SEM of 4 independent experiments, each including duplicate wells." There is one single data point on the graph with SEM – Is the datapoint the average of the 4 experiments, or are the 8 data points (4 experiments in duplicate) on the graph (is the SEM SqR of 4 or 8?), AND are the 4 experiments from 4 biologically distinct human donors, or the same donor measured four times.... None of this is clear and needs addressing. Similar issue in Figure 7.
- Similarly, some claims are made about significance, when there are no statistical tests

applied to the data. For example in Figure 7D, authors state “degranulation was significantly reduced, indicating the lower ICAM-1 availability reduces the capacity of cytotoxic T cells to activate LFA-1 and to secrete lytic granules”, yet there is no test applied to the data and the difference seems minimal (ie. 40% versus 50%).

- In addition to this, the methods states that: “statistical analyses used throughout the manuscript have been indicated in the figure legends”. However, tests are NOT listed in the legends, nor throughout the manuscript. This is a glaring omission and needs to be rectified.

- Figure 5 needs at least a representative video added to the supplementary files. Figure 5 and 6 show that increasing CD3 concentration (as a surrogate for TCR stimulation strength) associated with improved cytotoxicity. This figure is not novel, and has been shown for a long time and by many different groups and systems. The authors then state that this is due to “presumably mediated by LFA-1 activation determines the efficiency of the cytotoxicity activity”. It is worth pointing out that this is purely correlative and not definitive.

For example a study published in Blood in 2011 (PMID: 21123829) showed that increasing the concentration of anti-CD3 antibodies used to activate T cells in vitro led to higher levels of cytotoxicity against tumor cells. The study used human peripheral blood T cells and various cancer cell lines as targets. Another study published in Cancer Immunology, Immunotherapy in 2016 (PMID: 27465454) demonstrated that increasing the concentration of anti-CD3 antibodies used to activate T cells in vitro led to higher levels of cytotoxicity against melanoma cells. The study used human peripheral blood T cells and melanoma cell lines as targets. A study published in Cell Reports in 2018 (PMID: 30380413) used a transgenic mouse model to show that increasing the level of TCR expression on T cells led to improved cytotoxicity against tumor cells. The study also demonstrated that increasing TCR expression enhanced T cell activation and proliferation. Overall, these studies suggest that increasing the strength of TCR stimulation can enhance T cell cytotoxicity, and this effect has been observed in both in vitro and in vivo models. So while this present study is nice, it does lack some novelty and would require further evidence that the high affinity LFA-1 is contributing to the TCR tuning – the data presented is all correlative.

In this section below, I ask out of interest as a reviewer, rather than suggesting that these be experimentally addressed, however, reference to these concepts should be touched on in the Discussion.

-How do the authors think that the LFA-1 conformational change will be influenced when the T cells are not TCR stimulated, but rather via a CAR? What is happening inside the cell to promote the interplay between the CD3 stimulation and LFA-1.

-Similarly, while I really do understand the complexities of these experiments and using glass coverslips as targets for simplicity, have these experiments been done with TCR (like Tg OT-I) with diluted peptide presented on target cell membranes, in a cell-to-cell approach, and do authors see the same response or scaling LFA-1 open confirmation with enhanced stimulation?

-What role do the authors predict that CD8 is playing in the absence of MHC I. Have any of the experiments been conducted in the presence of anti-CD8 blocking antibody

-The low affinity antibody Hi-111 is directly conjugated, however, the high affinity antibody m24 recognizing the ‘open’ confirmation is biotin labelled and requires a secondary streptavidin binder, therefore increasing the stoichiometry of the labelling and improving the

signal of the 'open' binder. For transparency this needs to be acknowledged and the intensity of labelling can not be directly compared, and authors should be careful to acknowledge this when quantitating sizes of clusters for example (Figure 1).

In addition:

Small typos to be corrected:

Line 89: "specific of the" should read, Specific for the ...

Line 418 – Authors do not include the fluorophore for the Anti-CD107a antibody used (which they need to include) as this can severely impact the outcome, due to the quenching of pH instable fluorophores such as FITC.

Point-by-point reply to the reviewers

Our answers to each of the points raised by the reviewers appear in blue. Please note that we have added 2 figures with supportive preliminary data at the end of this document.

REVIEWER COMMENTS

Reviewer #1 (Remarks to the Author):

In this manuscript, Lacouture *et al.* use super resolution microscopy, together with conventional confocal and time-lapse microscopy, as well as flow cytometry to show that: (1) open and closed conformation LFA-1 appears in pre-formed nanoclusters on the surface of CD8 T cells; (2) T cell activation switches LFA-1 from closed to open conformation; (3) open conformation LFA-1 correlates with CD8 degranulation and killing and (4) that LFA-1 – ICAM1 interaction is required for efficient killing. While some findings confirm earlier work (e.g. the dependency of efficient killing on LFA1-ICAM1, the role of contact duration) the manuscript contains important findings about the nanoscale organisation of LFA-1 and the kinetic relationship between TCR signalling, integrin activation and degranulation.

We are glad to read that the reviewer found merit and novelty in our work.

Main points:

1) related to Figure 2: the main point of the authors here is that there appears to be a discrete 'pool' of LFA-1 nanoclusters that switches from closed to open conformation upon TCR stimulation. However, in line 136 (referring to Figure 2B) it is stated that the number of closed conformation LFA-1 clusters stays the same. Isn't this contradictory? If there is a finite pool the number of closed conformation LFA-1 clusters should go down while the number of open conformation should go up. Nevertheless, Figure 2C shows that the fraction of open conformation LFA-1 is going up. Doesn't this suggest that there is considerable *de novo* assembly of open conformation nanoclusters?

The reviewer is right that the fact that the rise in the number of open conformation nanoclusters is not mirrored by a corresponding drop in the number of closed conformation nanoclusters argues against a simple nanocluster conformation switch model, whereby a finite pool of nanoclusters would switch conformation. This point has also been raised by reviewer 2 (point 3). Although we considered the classical model pertaining to integrin activation (active clustering of isolated molecules) in the initial version of the manuscript, we did not discuss the apparent contradiction mentioned by the reviewers and positioned the nanocluster switch model probably too prominently.

To address these limitations, we have revised the text of the manuscript to provide a more balanced evaluation of the different conformation activation scenarios. Although it is beyond the scope of the work to fully elucidate the conformation activation mechanism, we have also worked at reinforcing the data pertaining to the preassembly of LFA-1 into closed conformation nanoclusters in non-activated T cells. We also completed our study by

evaluating the distribution of closed and open LFA-1 conformations at the synapse versus the rest of the cell, which provides a possible explanation for the raised contradiction.

In more detail, we have brought the following modifications to the revised manuscript:

1- In the section pertaining to Fig. 2, we have refocused the objective and interpretation of the data around the scalability of LFA-1 activation as a function of TCR stimulation. In particular we have removed the concluding sentence “Collectively, the study of the nanoscale distribution of LFA-1 conformations in CD8⁺ T cells supports the notion that pre-established LFA-1 nanoclusters operate as functional units of a TCR-driven digital activation process and that the LFA-1 nanocluster switch mechanism scales with TCR stimulation strength” and replaced it by “Collectively, these data reveal that the process of LFA-1 activation in the context of CD8⁺ T cell synapse assembly is sustained by the enrichment of open conformation nanoclusters at the stimulatory area, the number rather than the size of which grows with TCR stimulation strength” (lines 149-152).

2- In the discussion, we present the different conformation activation scenarios and address the apparent contradiction mentioned by the reviewers (see recomposed second chapter of the discussion, lines 314-341).

3- We carefully revised the text of the manuscript pertaining to the allusion to a nanoswitch mechanism, systematically explaining the still hypothetical nature of such a mechanism. This applies to the title of the manuscript that we have adjusted to align with the main findings of the work (scaling of LFA-1 activation at the nanoscale and cellular scale ; control of cytotoxicity by LFA-1 activation).

Pertaining to this point, we added the following new data to the revised manuscript:

1- Evidence for the pre-assembly of closed conformation LFA-1 nanoclusters in T cells that can truly be considered as non-activated. For that purpose, cells were stained in suspension to avoid that contact with glass or PLL non-specifically triggers LFA-1 reorganization (see Supplementary Fig. 2). Assessing this point was also motivated by point 1 from Reviewer 2.

2- Validation of the cluster recognition algorithm with artificial images in order to verify the reliability of the detection of closed conformation LFA-1 nanoclusters (see Supplementary Fig. 8). The Trainable Weka Segmentation algorithm, after being trained on *real* STED images, was able to distinguish artificial clusters from the background, in conditions comparable to those of the experiments.

3- Reinforcement of the evidence that closed and open LFA-1 conformations occupy distinct nanoclusters by a complementary SIM approach (see Supplementary Fig.1E-F).

4- Complementary measurements in T cell: APC conjugates were conducted by imaging flow cytometry in order to appreciate the distribution of closed and open conformations of LFA-1 at the synapse area versus the rest of the cell surface (see Fig. 3E-G). These data show that while open-conformation LFA-1 is mostly confined within the synapse, closed-conformation LFA-1 remains at equilibrium between the synapse and the rest of the cell. It implies that open-conformation LFA-1 is trapped within the synapse (presumably through its interaction with ICAM-1) and that closed-conformation LFA-1 diffuses from the membrane

outside the synapse to replenish the synaptic fraction that converts to open conformation, in virtue of thermodynamic equilibration between both cell membrane subsystems (synapse vs non-synapse membrane). Such diffusion-trapping process might explain the apparent contradiction highlighted by the reviewers that closed-conformation LFA-1 does not drop at the synapse when open-conformation LFA-1 increases. We could not solve whether such diffusion might be through isolated molecules or clusters. Anyway, this observation is relevant independently from the activation mechanism (de novo clustering vs switch of preassembled clusters), as discussed in the discussion section (lines 327-337).

The fact that we detected closed-state nanoclusters also on the dorsal side of cells forming synapses (not shown) may indicate that LFA-1 could also be recruited to the synapse area under this nanoscale organization.

Also related to Figure 2 and Figure S2: although the mean of the cellular area is staying constant, given the considerable spread of the data and the relatively low sample size (which is understandable given the used imaging technique), I would like to see these data normalized to the area of the cells (normalized number of clusters). Does this give cleaner data with a smaller standard deviation? This might also enable the authors to pool the data from their two independent experiments of which only one is shown.

The analysis of LFA-1 cluster densities per surface unit (μm^2) presented in Fig. 2D of the first version corresponds exactly to the normalization the reviewer is asking for. In the revised manuscript, we explain this analysis in a clearer way, by adding the sentence: "The number of detected LFA-1 nanoclusters was then normalized to the area of each of the studied cells" (line 143-144). We also explained this analysis in a clearer way in the corresponding figure legend. In order to better appreciate the distribution of the LFA-1 cluster density values across the considered cells, we opted for a new representation showing the data from the individual cells (Fig. 2D). This analysis indicates that the dispersion of individual values is not reduced (as compared to that of absolute LFA-1 cluster numbers per cell), implying that the cell to cell heterogeneity in the number of detected nanoclusters is not attributable to synapse area differences.

In order to reinforce our data, we present the data from the 2 original experiments done on T cells from one healthy donor and added those from a third experiment that we did with T cells from a second donor (Fig. 2B-D). Importantly we found that the range of values for the absolute cluster numbers and the cluster densities were matching across these experiments and therefore opted for displaying the pooled data. To reinforce our data, we also added a statistical analysis that clearly indicates that even if there is high dispersion across the analyzed cells, the numbers and densities of open-state LFA-1 nanoclusters increase upon TCR stimulation.

2) related to Figure 7: one experiment that might be interesting to do here is to vary the levels of ICAM1 on the target cells. The initial, relatively small differences despite full blockage of ICAM1 in Figure 7C & D might indicate that the system is quite robust against changes in ICAM1 levels.

As suggested by the reviewer, we attempted to vary the level of ICAM1 expression on the P815 target cells. We tested 6 different ShRNA, 5 yielding minor ShRNA knock-down and 1

yielding intermediate knock-down (data for the reviewer's attention presented at the end of this document in Fig. R1). The data indicate that when ICAM-1 was partially inhibited, LFA-1 activation and degranulation was partially affected. Minor reduction of ICAM1 had no measurable impact. These data are complementary to the ones with the inhibitory Ab in showing that the CD8-P815 killing is sensitive to ICAM-1 density but that residual killing is present even when ICAM-1 is inhibited.

3) P815 is a mouse cell line and thus expresses mouse ICAM-1 bound by the YN1/1 antibody (PMID: 2981916). This is fine as mouse ICAM-1 binds well to human LFA-1, but mouse LFA-1 expressed on P815 will bind poorly to human ICAM-1 (PMID: 2199576). P815 also lacks CD58, the other major adhesion ligand involved in CTL mediated killing with most human targets. Thus, this is actually a well-chosen target to isolate CTL LFA-1 to target ICAM-1 role in killing. These issues should be presented in explaining why P815 was chosen or discussed in relation to the point below; YN1/1 antibody will bind only to the P815 cells and not to ICAM1 on the human CTL. This is important because someone choosing a human target would probably get different results, even using antibodies to human ICAM-1.

We agree with the reviewer that we should point to the specificity of the LFA-1-ICAM-1 interaction across species, as important information to interpret our ICAM-1 blocking experiments in the context of the human T cells to murine P815 cell encounters. We have therefore edited the corresponding section in the revised version of the manuscript and added the suggested references (lines 287-291).

Please note that, as recommended by the two other reviewers, we completed this set of experiments by directly testing the contribution of the high-affinity conformation state to conjugate formation, degranulation and killing. To specifically block the conformation activation of LFA-1, we employed the BIRT 377 compound, a LFA-1-specific allosteric antagonist previously shown to block the induction the high-affinity conformation detected by the m24 antibody (Woska et al., J Leuk Biol 2001 ; Chigaev et al., Mol Biol Cell 2015). The results presented in the new Fig. 6 show that blocking the conformation activation of LFA-1 almost completely impaired lytic granule exocytosis, contacts with P815 cells and cytotoxic activity. This new piece of data is central to our study because it directly relates the process of LFA-1 conformation activation to the functionality of the cytotoxic T cells. The fact that the conformational inhibition of LFA-1 appeared to more severely impact lytic granule exocytosis than the blockade of ICAM-1-LFA-1 interaction via the blocking Ab YN1/1.7.4 suggests that P815 cells might trigger residual LFA-1 activation (and related degranulation) via receptors others than ICAM-1.

4) mAb 24 is a ligand induced binding site on LFA-1 beta subunit (PMID: 7685913). So the basal mAb 24 signal when there is no ICAM-1 on the substrate, or ICAM-1 on the target is blocked, is due to cis interaction with ICAM-1, which take place within LFA-1/ICAM-1 nanoclusters as documented on neutrophils (PMID: 30605669). Activated T cells express ICAM-1 that could engage LFA-1 in cis. It is possible that cis ICAM-1 interactions may participate in the mAb 24 binding within the activated nanoclusters. It would be helpful to cite the Ley lab paper and at least discuss its implications if not investigating whether the

activated LFA-1 clusters contain ICAM-1 or not. This should not be required, but I would be very interested to know the result!

We thank the reviewer for pointing to the work from the Ley lab on the binding of LFA-1 with ICAM-1 in cis. It is indeed possible that this mechanism, so far described in neutrophils, also applies to T cells. Along with the suggestion of the reviewer, we cite the work from the Ley lab and discuss the possibility that such a mechanism might explain the detection of m24 Ab signal in the absence of exogenous ICAM-1, but might also regulate the process of activation itself (see discussion of the revised manuscript, lines 336-338). We plan to investigate this point as part of a follow-up study.

5) Are you certain that mAb 24 + Hi-111 would be equal to all LFA-1, for example, as seen by TS2/4? Hi-111 has a 10-fold higher affinity for closed than open I domain, but mAb 24 is seeing a conformation that is physiologically generated by ligand in the beta I like domain, so there may be open I domain where the mAb 24 epitope is not generated as there was no ICAM-1 bound at the point of fixation. You are doing nice work with flow cytometry- can some book keeping be done with this approach to determine how many mAb 24 sites, Hi-111 and **TS2/4** (or some other pan LFA-1, non-blocking, epitope). This might help understand how both Hi-111 and mAb 24 sites can increase in some conditions. That may be related to recruitment of LFA-1 from other parts of the cell or due to some "stealth" LFA-1 that can give rise to both Hi-111 and mAb 24 positive under the correct conditions. Have the authors looked at the utility of **AL-57**, which binds active conformation of the I domain in a Mg²⁺ dependent manner.

We agree with the reviewer that the conformations of LFA-1 recognized by the m24 and Hi-111 antibodies are not expected to cover all LFA-1 conformations. We therefore revised our manuscript and replaced the term "total clusters" by "total detected clusters" in the figure panel (Supplementary Fig. 2A in initial version, now moved to Fig. 2B), in which we present the global number of detected clusters (m24 + Hi-111 Ab staining).

As suggested by the reviewer, we stained T cells with the pan-LFA-1 Ab TS2/4, expected to reveal all LFA-1 molecules independently from their conformation. Both dSTORM and confocal microscopy analysis indicate that TS2/4 staining distributes in part as nanoclusters, which display comparable size to the ones characterized with the m24 and HI-111 stainings. Quantification of the number of detected clusters of a limited number of cells suggests that TS2/4 stained clusters may exceed the sum of the m24 and HI-111 stained clusters. In agreement some of the clusters stained by the TS2/4 Ab appears negative for either of the Hi-111 or m24 Ab stainings (data for the reviewer's attention presented at the end of this document in Fig. R2).

If we understood well, the reviewer was also suggesting that we quantify LFA-1 binding sites to various Ab by cytometry. In the context of the new experiments we implemented to address the direct involvement of the high-affinity LFA-1 conformation in degranulation and cytotoxic activity, we tested combined stainings with the m24, Hi-111 and TS2/4 Abs followed by flow cytometry analysis (see new Fig. 6). However, because each antibody is bound to a different fluorochrome and read with a different laser, we could not compare the fluorescence intensities among the 3 conformations. To go further, one would wish to

quantify the absolute abundance of the different LFA-1 conformations. This is possible by flow cytometry using phycoerythrin-conjugated Abs and calibration beads. However, a limitation of such an approach is that we would then not be able to measure the relative abundance of each conformation on the same cells.

Reviewer #2 (Remarks to the Author):

Lacoutre et al. have studied the organization of the “closed”, i.e., low affinity and open, “high” affinity states of LFA-1 molecules, using STED microscopy and higher-throughput confocal microscopy-based imaging. The formation of LFA-1 nanoclusters has previously been reported in the context of T-cell migration (see Shannon et al., JCS, 2020), but the relationship between LFA-1 nanoclusters and conformational changes has not been addressed, especially in the context of T-cell synapse formation. Lacoutre et al. propose that LFA-1 forms distinct nanoclusters containing only one or the other states, that the number of open-state nanoclusters scales with the level of TCR signaling and, importantly, that this involves a concerted “switch-like” mechanism wherein a given closed-state nanocluster converts en bloc to an open state. They go on to suggest that this switch conditions T-cell/target contact stability, lytic granule exocytosis, and killing. The experiments are thought-provoking and, for the most part, well performed and the paper is well written. Nevertheless, I have important reservations concerning the methods used and the interpretation of the data, that undermines my confidence in the significance of the findings. Overall, the authors’ claim that a switch-like opening of LFA-1 nanoclusters licenses killing of target cells is far from being convincingly demonstrated.

We fully understand that the reviewer is challenging the interpretation of our data pertaining to the conformational activation of LFA-1 at the nanoscale. As explained below in our answers, the points raised by the reviewer have stimulated us to complete and consolidate our study as well as to partially revise some of our initial statements, in particular regarding the “en bloc conversion” model.

We thank the reviewer for pointing to the study by Shannon et al., which we have added to our introduction, together with the other studies we had cited as relevant for the organization of LFA-1 into nanoclusters.

1) First, the imaging experiments were done on glass, which is problematic insofar as glass has been shown to be capable of activating T cells, and poly L-lysine even more so (Santos et al. NI, 2018). So, the issues then are: what does the true resting state look like, and is it safe to conclude that LFA-1 is pre-clustered in the resting state at all? It could well be that only very small numbers of closed-state nanoclusters, if any, exist in the truly resting state, and so the notion that activation is associated with the conversion of existing nanoclusters would be incorrect. The authors refer to work by others claiming that LFA-1 is pre-clustered, for example Baumgart et al., but Baumgart et al. also worked from glass surfaces. To establish the starting organization, it would be necessary to keep the cells free from the influence of all surfaces. At the very least the distribution in the upper surface of a cell would be worth determining.

In order to best approach a “true resting state”, we sought to perform STED analysis following staining and fixation of the T cells in suspension before they were deposited on uncoated glass slides. The new data presented in Supplementary Fig. 2A-B show that the closed conformation of LFA-1, as revealed by the Hi-111 Ab staining, is at least in part assembled in nanoclusters comparable to those characterized across the different stimulatory conditions. The number of detected nanoclusters per surface unit in the resting T cells was comparable to that detected in the T cells deposited on PLL (approximately 3

nanoclusters per μm^2), suggesting that the seeding of T cells on PLL coated glass did not bias the nanoscale organization of LFA-1. Note that we also examined the Hi-111 Ab staining in the upper surface (dorsal side) of T cells deposited on various surfaces and observed a comparable distribution of closed conformation LFA-1 (data not shown).

2) Second, notwithstanding the above, I wasn't wholly convinced that the STED data actually supports the notion of clustered closed-state LFA-1 in Fig. 1A. Certainly, the segmentation algorithm produces nanoclusters, but there seems to me to be a step difference in the degree of local intensity variation for the closed- and open-state staining, and in the area of the cell surface across which the intensity varies. Without controls, it's not clear that the closed-state nanoclusters picked by the algorithm aren't artefactual. What's needed here would be, e.g., a membrane stain, or some other protein not thought to be clustered. The Baumgart et al. paper could be used to identify suitable controls and it would be helpful to the authors if the algorithm failed to cluster these. It would also have been helpful to have performed calibration experiments to relate signal intensity to protein number, e.g., by performing STED on low densities of the antibodies adsorbed to glass, so that it can be certain that clusters are really forming. It's also unclear from the methods whether the diffuse non-clustered LFA-1 signal truly reflects monomeric protein or non-specific binding of the antibody, as the authors do not indicate whether the cells were washed prior to fixation/imaging.

We agree with the reviewer that closed-state LFA-1 appeared to cluster less clearly than open-state LFA-1, at least on the basis of the examination of the raw STED images (Fig. 1A). This is actually reflected by the quantification of the proportion of staining in the detected clusters (Supplementary Fig 1A) showing that less Hi-111 Ab staining is associated to clusters than what is the case for m24 Ab staining. In order to ascertain that a proportion of closed-state LFA-1 is assembled in nanoclusters, we worked along 3 lines, as detailed below: i) we estimated the probability that observed clusters correspond to local density fluctuations of randomly dispersed molecules ; ii) we validated our nanocluster segmentation algorithm with *in silico* generated images ; iii) we complemented our STED analysis by applying complementary super-resolution microscopy modalities to the detection of LFA-1 nanoclusters.

As explained in the first chapter of the supplementary material, we modeled the synapse to estimate the probability that what we detected as closed-state nanoclusters by STED imaging would correspond to local density fluctuations of randomly dispersed LFA-1 molecules. Our modelling approach indicates that even when setting unfavorable parameter values for density of LFA-1 molecules on the plasma membrane and density of LFA-1 molecules in the clusters, it is extremely unlikely that the detected clusters correspond to random density fluctuations of an otherwise homogeneous distribution of molecules.

In order to validate the nanocluster segmentation algorithm, we sought that the suggestion of the reviewer to do a simulation (minor point 4) was very relevant. We therefore generated test images *in silico* (see supplementary material and Supplementary Fig 8). The algorithm trained on *real* STED images was very efficient in picking up the nanoclusters and to distinguish them from the background. Although it tended to pick up a few additional clusters

from the background (false positives), those remained a minor fraction of the detected clusters.

The initial version of the manuscript was already containing data pertaining to the analysis of closed-state LFA-1 distribution by dSTORM (Supplementary Fig. 1C). The reconstructed images and the DBSCAN analysis are indicating that closed-state LFA-1 is at least in part assembled into nanoclusters in activated T cells. Furthermore the detected nanoclusters have a typical size comparable to that of the open-state nanoclusters. This is in line with a previous report that compared the distribution of different LFA-1 conformations in migrating T cells by dSTORM (Persson *et al.*, J Biophotonics 2019).

We completed this complementary analysis using SIM (Supplementary Fig. 1E-F). Although SIM does not offer a spatial resolution as important as dSTORM and STED, it was sufficient to discriminate single-state LFA-1 nanoclusters, thereby confirming that LFA-1 forms nanoclusters independently from its conformation and that these nanoclusters tend to be well segregated one from another (would not be distinguishable by SIM otherwise). By applying a simple colocalization analysis to SIM images independently from any cluster segmentation, we also confirmed that the closed and open conformations of LFA-1 did not overlap, indicating that a conformation state is coordinated locally at the scale of the nanoclusters.

Together, our modeling and simulation approaches, as well as the new SIM data, reinforce the notion that at least part of the closed-state LFA-1 molecules assemble in nanoclusters at the surface of human CD8⁺ T cells.

3) Third, and perhaps most importantly, I was unconvinced by the central claim of the paper, which is that there's "digital" conversion of closed-state nanoclusters to open-state ones. All things being equal, if there was digital conversion of closed-state nanoclusters to open-state ones, the number of closed-state nanoclusters would decrease in proportion to the increase in open-state nanoclusters, but this is not what is seen: both increase, albeit slowly in the case of the closed-state LFA-1. I think that, considering also the data in Fig. S5, it's more likely that there is an increase in open-state nanoclusters, formed from the pool of unclustered closed-state LFA-1. This would fit with the conventional view of cluster formation.

The reviewer is right that the fact that the rise in the number of open-state nanoclusters is not mirrored by a corresponding drop in the number of closed-state nanoclusters argues against a simple nanocluster conformation switch model, whereby a finite pool of nanoclusters would switch conformation. This point has also been raised by reviewer 1 (point 1). Although we considered the classical model pertaining to integrin activation (active clustering of isolated molecules) in the initial version of the manuscript, we did not discuss the apparent contradiction mentioned by the reviewers and positioned the nanocluster switch model probably too prominently.

To address these limitations, we have revised the text of the manuscript to provide a more balanced evaluation of the different conformation activation scenarios. Although it is beyond the scope of the work to fully elucidate the conformation activation mechanism, we have also worked at reinforcing the data pertaining to the preassembly of LFA-1 into closed

conformation nanoclusters in non-activated T cells. We also completed our study by evaluating the distribution of closed and open LFA-1 conformations at the synapse versus the rest of the cell, which provides a possible explanation for the raised contradiction.

In more detail, we have brought the following modifications to the revised manuscript:

1- In the section pertaining to Fig. 2, we have refocused the objective and interpretation of the data around the scalability of LFA-1 activation as a function of TCR stimulation. In particular we have removed the concluding sentence “Collectively, the study of the nanoscale distribution of LFA-1 conformations in CD8⁺ T cells supports the notion that pre-established LFA-1 nanoclusters operate as functional units of a TCR-driven digital activation process and that the LFA-1 nanocluster switch mechanism scales with TCR stimulation strength” and replaced it by “Collectively, these data reveal that the process of LFA-1 activation in the context of CD8⁺ T cell synapse assembly is sustained by the enrichment of open conformation nanoclusters at the stimulatory area, the number rather than the size of which grows with TCR stimulation strength” (lines 149-152).

2- In the discussion, we present the different conformation activation scenarios and address the apparent contradiction mentioned by the reviewers (see recomposed second chapter of the discussion, lines 314-341).

3- We carefully revised the text of the manuscript pertaining to the allusion to a nanoswitch mechanism, systematically explaining the still hypothetical nature of such a mechanism. This applies to the title of the manuscript that we have adjusted to align with the main findings of the work (scaling of LFA-1 activation at the nanoscale and cellular scale ; control of cytotoxicity by LFA-1 activation).

4- Complementary measurements in T cell: APC conjugates were conducted by imaging flow cytometry in order to appreciate the distribution of closed and open conformations of LFA-1 at the synapse area versus the rest of the cell surface (see Fig. 3E-G). These data show that while open-conformation LFA-1 is mostly confined within the synapse, closed-conformation LFA-1 remains at equilibrium between the synapse and the rest of the cell. It implies that open-conformation LFA-1 is trapped within the synapse (presumably through its interaction with ICAM-1) and that closed-conformation LFA-1 diffuses from the membrane outside the synapse to replenish the synaptic fraction that converts to open conformation, in virtue of thermodynamic equilibration between both cell membrane subsystems (synapse vs non-synapse membrane). Such diffusion-trapping process might explain the apparent contradiction highlighted by the reviewers that closed-conformation LFA-1 does not drop at the synapse when open-conformation LFA-1 increases. We could not solve whether such diffusion might be through isolated molecules or clusters. Anyway, this observation is relevant independently from the activation mechanism (de novo clustering vs switch of preassembled clusters), as discussed in the discussion section (lines 327-337).

4) Fourth, in my view the authors fail to establish a strong link between adhesion or killing and the conversion of nanoclusters from closed to open, if it is assumed that this occurs. It must be said that the correlation between the gradual changes observed using STED (Fig. 2D) marking nanocluster “conversion”, and the sigmoidal responses observed, e.g., for degranulation (Fig. 4B) and, especially, doublet formation (Fig. S4B), with anti-CD3

concentration is poor. The correlations between the overall levels of each LFA-1 state measured confocally and, e.g., degranulation versus anti-CD3 concentration is far more convincing. The problem, of course, is that it's impossible to know which form of LFA-1 is producing the key effects in the functional assays (assuming the measurements on glass are meaningful). Is it the relatively small numbers (20%) of LFA-1 proteins in nanoclusters or the large number of unclustered molecules. These issues are compounded by the fact that attempts weren't made to determine whether the cells that were imaged using STED had actually triggered, and by only a single time-point (20 min) being examined, leaving uncertainty about the true dynamics of nanocluster "conversion" and its real dependence on TCR signaling, on a per cell basis.

The reviewer is right that the initial version of our manuscript suffered from not providing a direct link between adhesion/killing and LFA-1 conformation activation. The correlative nature of the data we were providing to sustain such a link was also raised as a limitation of our work by reviewer 3 (points 3 and 7). In addition to our initial data, it might be assumed that LFA-1 conformation activation is required for killing because the LFA-1-ICAM-1 interaction is required for optimal killing and the killing process is accompanied by LFA-1 conformational activation. However, it remains that the requirement of the high-affinity conversion to sustain killing has not been tested directly. To fill this gap, we specifically blocked the conformation activation of LFA-1, employing the BIRT 377 compound, a LFA-1-specific allosteric antagonist previously shown to block the induction of the high-affinity conformation detected by the m24 antibody (Woska et al., J Leuk Biol 2001 ; Chigaev et al., Mol Biol Cell 2015). The flow cytometry results presented in the new Fig. 6 show that blocking the conformation activation of LFA-1 (without impacting the expression of LFA-1 itself, as verified with the Hi-111 and TS2/4 Abs) almost completely impaired lytic granule exocytosis, contacts with P815 cells and cytotoxic activity. This new data is central to our study because it directly relates the process of LFA-1 conformation activation to the functionality of the cytotoxic T cells.

We believe that further elucidating the relative contribution of the open-state nanoclusters and the more isolated open-state molecules to the killing process is beyond the scope of this work. Along with the comments of the reviewers, we can mention that our preliminary data on live cell imaging indicate that the process of LFA-1 activation is very fast and precedes the detection of killing. We have not yet monitored TCR signaling (e.g. inside-out signaling) during the process of LFA-1 activation and killing on a single-cell basis but will consider this suggestion for our follow-up work.

5) Finally, I was struck by the very small effect of the anti-CD3 antibody on the ratio of closed/open state LFA-1 measured confocally in Fig. S5, in the absence of ICAM-1 (but also in the presence). This looks like an ~10% shift. What does this say about inside-out signaling? Doesn't this undermine the thesis that the TCR oversees LFA-1?

The concept that TCR engagement and resulting inside-out signaling is required for the conformational activation of LFA-1 to the high-affinity state in the context of the T cell synapse is well established. Our work provides novel insight to this concept by examining at

different biological scales how LFA-1 activation is tuned upon TCR engagement. Our data acquired via STED imaging, high-content confocal imaging and flow cytometry largely confirm that LFA-1 activation depends on TCR engagement. Regarding specifically the data initially presented in Figure S5 (based upon the same data as Fig. 7A-B), we believe the minor shift in the closed/open state ratio is related to the relative overweight of the closed state conformation and the fact that the closed state conformation also increases (even if much less than the open state conformation). Furthermore, when compared to the value with vanishing anti-CD3 concentration, the increase is more pronounced, consistently with the fact that the main LFA-1 activation effect has occurred below the concentration of 0.1 $\mu\text{g/mL}$ (see new Figures 2B and 4B). We understand that this piece of analysis might be misleading and removed it from the revised manuscript, all the more so since it does not contain new information as compared to previous figures.

Minor points/questions

1. Line 56: what's meant by "metrics of TCR/LFA-1 cooperation"?

"Metrics" has been replaced by "quantification".

2. Line 80: what's meant by "discretization of signals into nanoswitches"? What is "discretization".

We employed the term "discretization" used in mathematics (where "discrete" is the opposite of "continuous") to convey the notion that the distribution of LFA-1 corresponds to a finite number of discrete nanoclusters rather than dispersed individual molecules. We propose that a digital activation of the approx. 1000 clusters detected at the synapse allows for very graded adaptative response. Such discretization also implies that LFA-1 nanoclusters act as functional units for their adhesive and mechanotransduction function.

3. Line 94: "pictures" should be "images".

"Pictures" has been replaced by "images".

4. Line 96: if there's only three-fold enrichment of LFA-1 in the 80-90 nm diameter clustered areas, what is it that qualifies as a nanocluster (also see comments above about the stoichiometry of the clusters)? How would a random distribution look according to the segmentation algorithm? Would a simulation be helpful?

This comment from the reviewer stimulated a new analytical development aiming at verifying the calibration of our cluster segmentation algorithm (see answer to point 2, above).

5. Line 101: what's driving the formation of closed nanoclusters (if they form)?

As explained in our answer to point 2, we have worked at providing additional evidence for the presence of closed-state LFA-1 nanoclusters at the surface of human CD8⁺ T cells. The chemical/physical mechanisms underlying the assembly of closed-state and open-state LFA-1 nanoclusters have not been elucidated in this study. However, we have now added a chapter in the discussion that presents different scenarios (lines 310-341). In particular we have referred to [N. Destainville, M. Manghi, J. Cornet, A rationale for mesoscopic domain formation in biomembranes, *Biomolecules* 8, 104 (2018)].

6. Line 168: rather than “cells with higher surface LFA-1 expression are more prone to activate it” it’s clear that the ratio of closed/open states is constant for all levels of LFA-1 expression in Fig. 3C, suggesting that the cells are equally prone to activate the same fraction of LFA-1, irrespective of expression level; it’s just that there’s more of it for the higher-expressing cells.

The reviewer is absolutely right. We have therefore replaced “cells with higher surface LFA-1 expression are more prone to activate it “ by “LFA-1 activation scales with the level of available LFA-1 at the surface of individual cells” (lines 169-170).

7. Line 328: the use of “smoothly” is inappropriate.

Smoothness is again a mathematical terminology, meaning that a function is regular, without singularities or discontinuities, like a “step”. It is not fully appropriate indeed, and was only intended to illustrate the more regular character as compared to a sigmoid. We have removed it since it did not add much to the term “gradually” positioned in the same sentence (line 376).

8. Line 441: the HCS acronym should be defined.

“HCS device” was replaced by “high-content confocal microscope”

9. Line 441-442: what temperature and CO₂ level was used?

“equipped with temperature and CO₂ control” was replaced by “set to 37°C and 5% CO₂.”

10. Fig. 1: have the (bivalent) antibodies altered the clustering of LFA-1? Would the results be similar if the LFA-1 was labelled with a genetically-encoded tag or using Fab fragments?

Whereas the m24 Ab is fixation sensitive and needs to be added for at least a few minutes to the cells prior to fixation, the other LFA-1 specific Ab are not. In our hands staining with Hi-111 Ab prior to or after fixation yields comparable results in terms of nanocluster distribution and size. We are therefore confident that this antibody does not alter the clustering of LFA-1. Furthermore, Cambi et al., (MBOC 2006) also used antibodies against the alpha chain of LFA-1 (like Hi-111) for electronic microscopy and reported LFA-1 nanoclusters in monocytes, but random distribution in immature dendritic cells. The observation of random

distribution in at least one cell type support the notion that the assembly of nanoclusters is not an artefact of the staining with bivalent antibodies.

Our data are also aligning with those of previous analysis of LFA-1 distribution by super-resolution microscopy approaches in which the staining is clearly indicated as having being performed after fixation (Shannon et al, J Cell Science 2019, clone 2D7 used on fixed murine T cells stimulated over ICAM-1). None of the previous analysis of LFA-1 distribution by super-resolution microscopy approaches we are aware of (Baumgart et al, Nat Methods 2016 ; Murugesan et al., J Cell Biol 2016 ; Houmadi et al., Cell Rep 2018 ; Persson et al, J Biophotonics 2019; Shannon et al, J Cell Science 2019) reports the use of a genetically-encoded tag or Fab fragments. We believe Fab fragments are well suited also because of their reduced size limiting the decoration of the studied clusters with too large staining molecules. We have attempted to generate Fab fragments from commercial lots of LFA-1 Ab and to couple them to fluorophores but failed to generate enough good quality staining reagents. We understand that such approach would necessitate access to relatively large batches of antibodies as produced from hybridomas.

11. Fig. 2: it would be helpful to know the contribution of inside-out signaling vs. ICAM-1 engagement on driving the arrangements observed, by comparing open conformation nanoclustering on glass with and without ICAM-1.

The data presented in revised Fig. 2 allow us now to compare the nanoscale topography of the different conformations of LFA-1 in resting T cells and T cells stimulated by ICAM-1 only. Spreading on ICAM-1 appears to allow a first level of LFA-1 conformation switch. The open-state nanoclusters we detected tend to distribute at the mid-zone to rear of the polarized cells, in agreement with the work from the Hogg lab (Stanley et al., EMBO J 2008).

12. Figs 4, 5: a control with no LFA-1 engagement/blocking LFA-1 ought to be included in these experiments. Elsewhere it has been shown that ICAM-1 is dispensable for T-cell mediated killing of tumour cells (see Regev et al. Frontiers Immunol, 2022).

Fig. 4B-C actually contain a control without LFA-1 engagement (T cells alone) aiming at measure the basal level of open-state LFA-1 and LAMP1 expression at the surface of the cytotoxic T cells. Such control was not relevant for Figure 5, which is focused on T cell to target cell interaction events. In this context, we did test the effect of ICAM-1 blockade (Figure 7), ICAM-1 knock-down (see answer to point 2 of Reviewer 1 and Fig. R1 at the end of this document), as well as LFA-1 conformational activation blockade (new Fig. 6).

13. Fig. 4A: I think the second row should start with a concentration label of 0.1 µg/ml.

This is correct (has been corrected).

14. Fig. 4c: how was the threshold set for analysing LAMP+ cells for the MFI measurements shown? This should be indicated as a supplementary figure or in the methods.

It is the gate shown in Fig. 4A (superior right and inferior right parts of the quadrant). This has been better explained in the corresponding legend.

15. Fig. S4D: what are we to make of the 60% of LAMP+ cells that are neither m24+ or m24-? Surely, the fraction of cells that are either positive or negative for any given label is 100%.

The Y axis in this figure represents the entire cell population. We have made this clearer in the new version of the manuscript by indicating "% of total cells" for the Y axis label.

16. From Fig. S5 it appears that the presence of ICAM-1 has only very modest effects on the ratio of open to closed LFA-1. Is ICAM-1 being engaged properly on the surface? When adsorbed to glass it will likely adopt multiple positions, some more/less accessible than others to LFA-1. This possibility should be acknowledged.

For the reasons mentioned in point 5, we have removed Supplementary Fig. 5 from the revised manuscript. Regarding the question of ICAM-1 accessibility that is also relevant for numerous other figures of the work, we have titrated the concentration of Fc-tagged recombinant human ICAM-1 in a functional assay based on the adhesion of human CD8⁺ T cells. The concentration of 2 µg/mL used in this work was chosen because it corresponds to the beginning of the plateau of the adhesion response curve. We opted for the use of ICAM-1 fused to a Fc fragment for the exact point of the reviewer regarding its orientation during adsorption on glass or plastic. However, we cannot exclude that some of the coated molecules are not properly oriented to engage with LFA-1.

17. More experimental detail should have been included in the Methods. Were the cells ever washed after staining with OKT3 (lines 416/417, 426/427)? OKT3 in solution can block further OKT3 binding and, potentially, induce signaling. Simply stating that a 2:1 ratio of cells was used (line 428 and 438) is insufficient, when cell density could impact the dynamic range of cell conjugation/killing experiments; actual numbers and densities should be noted. What was the concentration of CellTrace violet used (line 437)? Also, it's indicated that contact time, contact number and lethal cytotoxicity were manually quantified (line 431-432). But how were the cell contacts detected/defined? Was this using brightfield? More details are needed to appropriately judge whether manual analysis was a reasonable approach or not.

We edited the Method section to provide the experimental detail asked by the reviewer.

Reviewer #3 (Remarks to the Author):

It has been long established that T cell clustering represents a mechanism by which T cell differentiation and effector function is controlled. Some studies have shown that LFA-1 undergoes conformational changes upon binding to ICAM-1, leading to the formation of stable adhesion complexes between T cells and target cells. The strength of the adhesion can affect the sensitivity of T cells to antigen stimulation, as well as the duration and intensity of T cell activation. In addition, the density of ICAM-1 on target cells can influence the degree of LFA-1 clustering and signalling, which can impact T cell activation and killing.

One study published in the Journal of Immunology in 2015 (PMID: 26324742) found that the strength of T cell receptor signalling was positively correlated with the density of ICAM-1 on target cells and that LFA-1 played a critical role in this process. Another study published in Immunity in 2019 (PMID: 30893595) showed that LFA-1 signalling could be modulated by the strength of T cell receptor signalling, and that this modulation was important for regulating T cell activation and killing.

In this manuscript, the Lacouture et al, provide evidence that LFA-1 confirmation activation scales with TCR stimulation strength to calibrate killing of target cells, and that LFA-1 'licenses' rather than controls the killing decision. The results are very interesting and will be of interest to the field.

Overall, it is an interesting study, and used lovely microscopy to tease apart the role of LFA-1 in T cell function. However, I do have some questions as to the robustness of the data (ie. Only analyzing 14 cells) and the interpretation, given the models used in this system were non-physiological stimulation (OKT3), rather than TCR-mediated, and controlling for affinity and role of other molecules such as CD8.

We thank the reviewer for the interest in our study. We felt sorry for not finding the two references the reviewer mentioned regarding the interplay between TCR signaling and LFA-1 activation (search for the indicated PMID numbers was fruitless). We would be eager to add them to our introduction if the reviewer could provide more details.

Here are some of the issues that require addressing in the manuscript:

1) ICAM availability on the target cells is a crucial component. Can the authors also justify the choice of using ICAM-1 conc.– how does this concentration compare the physiological levels – Is it saturating?

As shown in Fig. R1A at the end of this document, P815 target cells express homogeneous and high levels of surface ICAM-1. In order to complement the data pertaining to the blockade of the ICAM-1 to LFA-1 interaction with a specific Ab in the context of the T cell to target cell conjugates (as presented in Fig. 7), we attempted to reduce the level of ICAM1 expression on the P815 target cells with a ShRNA approach (Fig. R1A). Our data indicate that only the most efficient knock-down (drop of ICAM-1 expression to intermediate levels) was associated with a functional impact on LFA-1 activation and degranulation. The high ICAM-1 expression on P815 cells might therefore be saturating in terms of its ability to

interact with LFA-1 at the surface of human effector T cells. In line with this consideration, the concentration of 2 $\mu\text{g}/\text{mL}$ used in this work to stimulate 2D synapse assembly (in the absence of target cells) was chosen because it corresponds to the beginning of the plateau of the adhesion response curve of human CD8^+ T cells.

2) The authors perform a lovely STED analysis to show that open LFA-1 nanoclusters scale with TCR stimulation strength. In Figure 2 and S2 – It is not clear in Figure 2B whether the zero CD3 concentration, have also need stimulated with ICAM alone, as per Figure S2, or whether this group is just T cells on glass with no PLL, ICAM, or CD3. It is very hard to follow and requires further labelling, or detail in the figure legends. The methods seem to indicate that the zero concentration of CD3, also has ICAM present at 2 $\mu\text{g}/\text{mL}$ and if this is the case, this should be stated in the legend. Where there statistical tests applied to the difference in cluster number between the PLL alone, and ICAM as they look statistically different. And if so, would the authors expect this? These types of analyses should really be performed on several human donors, and on more than 14 cells, to enable robust statistical power to be obtained. Did the authors conduct a power calculation and can justify why they believe only 14 cells was sufficient to draw conclusions?

We apologize if the initial version of our figures were not clear enough. We have systematically relabelled the plots from Fig. 2 and Supplementary Fig. 2 to clarify which coating was used in each condition. We would like to draw the attention of the reviewer that our initial data did not include a true resting state in which T cells would not react to glass or PLL. As suggested by reviewer 2, we have explored the distribution of LFA-1 also in T cells in suspension that were stained before fixation and loading on glass slides. These new data are presented in Supplementary Fig. 2A-B.

We followed the advice of the reviewer to provide statistical analysis in order to test the difference in cluster number between the PLL and ICAM-1 condition. As shown in the new version of Fig. 2 the total number of detected clusters indeed increases and this remains true when it is normalized to take into account the increase of cell spreading elicited by ICAM-1. Our explanation is that the presence of ICAM-1 "traps" LFA-1 clusters (mostly in the closed-state conformation as shown in Fig. 2C-D). The detection of open-state clusters in the ICAM-1 condition is expected if we consider the seminal work from the Hogg lab (Stanley et al., EMBO J 2008). Indeed, spreading on ICAM-1 appears to allow a first level of LFA-1 conformation activation that mostly distributes at the mid-zone to the rear of the polarized T cells.

The remark of the reviewer about the fact our initial manuscript was presenting data stemming from T cells of a single healthy donor is well taken. During the revision of the work, we have invested in conducting confirmation experiments to reinforce the LFA-1 nanocluster quantification data. Fig. 2B-D and Supplementary Fig. 2C now present the pooled results of 3 independent experiments conducted on T cells from 2 donors. As indicated in the corresponding legend, a total of 27, 38, 40, 42, 41, 42, and 39 cells were studied per stimulatory condition, respectively for PLL, 0, 0.1, 0.5, 1, 5 and 10 $\mu\text{g}/\text{mL}$ anti-CD3 Ab. Collecting data from such number of cells allowed us to conduct a proper statistical analysis that clearly indicates the impact of ICAM-1 and TCR stimulation on cluster numbers and densities.

3) Have the authors considered blocking the conformational change of LFA-1 using either peptide inhibitors, monoclonal antibodies, or small molecule inhibitors? This could be a critical element for this study to improve impact and definitely show that it is the LFA-1 change that is tuning the TCR.

We understand that the reviewer is suggesting that we directly test the role of LFA-1 conformational change in tuning the TCR-driven functional outcome (degranulation, killing). This is an absolutely central question and we are grateful to the reviewer to raise this. We recognize that the initial version of our manuscript suffered from not providing a direct link between adhesion/killing and LFA-1 conformation activation. The correlative nature of the data we were providing to sustain such a link was also raised as a limitation of our work by reviewer 2 (point 4). In addition to our initial data, it might be assumed that LFA-1 conformation activation is required for killing because the LFA-1-ICAM-1 interaction is required for optimal killing and the killing process is accompanied by LFA-1 conformational activation. However, it remains that the requirement of the high-affinity conversion to sustain killing has not been tested directly.

To fill this gap, we specifically blocked the conformation activation of LFA-1, employing the BIRT 377 compound, a LFA-1-specific allosteric antagonist previously shown to block the induction of the high-affinity conformation detected by the m24 antibody (Woska et al., J Leuk Biol 2001; Chigaev et al., Mol Biol Cell 2015). The flow cytometry results presented in the new Fig. 6 show that blocking the conformation activation of LFA-1 (without impacting the expression of LFA-1 itself, as verified with the Hi-111 and TS2/4 Abs) almost completely impaired lytic granule exocytosis, contacts with P815 cells and cytotoxic activity. This new data is central to our study because it directly relates the process of LFA-1 conformation activation to the functionality of the cytotoxic T cells.

4) There is not sufficient detail in figure legends to determine exactly what data points are on the graphs and needs some careful attention. For example, Figure 3B authors state, “Mean and SEM of 4 independent experiments, each including duplicate wells.” There is one single data point on the graph with SEM – Is the datapoint the average of the 4 experiments, or are the 8 data points (4 experiments in duplicate) on the graph (is the SEM SqR of 4 or 8?), AND are the 4 experiments from 4 biologically distinct human donors, or the same donor measured four times... None of this is clear and needs addressing. Similar issue in Figure 7.

We have revised the figure legends to more precisely indicate the number of cell analyzed, the number of technical replicates, the number of independent experiments and the number of donors from whom the T cells were studied. We also better explained what the symbols on the graphs exactly represent.

5) Similarly, some claims are made about significance, when there are no statistical tests applied to the data. For example in Figure 7D, authors state “degranulation was significantly reduced, indicating the lower ICAM-1 availability reduces the capacity of cytotoxic T cells to activate LFA-1 and to secrete lytic granules”, yet there is no test applied to the data and the difference seems minimal (ie. 40% versus 50%).

As highlighted by the reviewer we recognize that the first version of the paper lacked significance tests. Following the advice of the reviewer, we consolidated the data with new experiments (e.g. in Fig. 2 and Supplementary Fig. 2), and applied statistical analysis to test their significance. The legends were edited in order to indicate which tests were applied to each dataset. All together, we believe that the statistical analysis we applied throughout the presented data very much reinforces the main messages of the work.

In particular for Fig. 7D, if we compare the percentage of LAMP1⁺ cells of [Isotype treatment vs YN1/1.7.4 treatment] at 10 µg/mL anti-CD3 Ab, we measured [67.3 vs 46.1] for the first experiment, [56.5 vs 38.9] for the second experiment and [44.04 vs 28.4] for the third experiment. This corresponds to a decrease of more than 30% of the degranulation capacity of the T cells. The statistical significance of this observation was verified by ratio paired t-tests applied at different anti-CD3 Ab concentrations, as indicated in the new version of the figure.

6) In addition to this, the methods states that: “statistical analyses used throughout the manuscript have been indicated in the figure legends”. However, tests are NOT listed in the legends, nor throughout the manuscript. This is a glaring omission and needs to be rectified.

As discussed above in our response to point 5, we have worked at reinforcing the manuscript by providing adequate statistical analysis. The legends were edited in order to indicate which tests were applied to each dataset.

7) Figure 5 needs at least a representative video added to the supplementary files.

Figure 5 and 6 show that increasing CD3 concentration (as a surrogate for TCR stimulation strength) associated with improved cytotoxicity. This figure is not novel, and has been shown for a long time and by many different groups and systems. The authors then state that this is due to “presumably mediated by LFA-1 activation determines the efficiency of the cytotoxicity activity”. It is worth pointing out that this is purely correlative and not definitive.

For example a study published in Blood in 2011 (PMID: 21123829) showed that increasing the concentration of anti-CD3 antibodies used to activate T cells in vitro led to higher levels of cytotoxicity against tumor cells. The study used human peripheral blood T cells and various cancer cell lines as targets. Another study published in Cancer Immunology, Immunotherapy in 2016 (PMID: 27465454) demonstrated that increasing the concentration of anti-CD3 antibodies used to activate T cells in vitro led to higher levels of cytotoxicity against melanoma cells. The study used human peripheral blood T cells and melanoma cell lines as targets. A study published in Cell Reports in 2018 (PMID: 30380413) used a transgenic mouse model to show that increasing the level of TCR expression on T cells led to improved cytotoxicity against tumor cells. The study also demonstrated that increasing TCR expression enhanced T cell activation and proliferation. Overall, these studies suggest that increasing the strength of TCR stimulation can enhance T cell cytotoxicity, and this effect has been observed in both in vitro and in vivo models. So while this present study is nice, it does lack some novelty and would require further evidence that the high affinity LFA-1 is contributing to the TCR tuning – the data presented is all correlative.

As suggested by the reviewer, we have added a representative movie (Supplementary Movie 1) as supplementary material to illustrate the setting used to produce the data shown in Fig. 5A-E.

We agree with the reviewer that the fact that increasing anti-CD3 Ab concentration (as a surrogate for TCR stimulation strength) is associated with improved cytotoxicity is not a novel data per se. We could however not identify the references mentioned by the reviewer. As the reviewer understood, our original point in this manuscript is about the role of LFA-1 in mediating the TCR-driven tuning of cytotoxic activity. We agree that in the initial version of our manuscript our assumption (LFA-1 conformation activation mediates TCR-driven tuning of cytotoxic activity) was only based on a set of correlations and overlay of response curves. As already discussed in point 3, the reviewer made us aware that providing direct evidence for such mechanism is central to our work. As explained in point 3 our strategy during the revision has been to challenge this question with the BIRT 377 compound, a LFA-1-specific allosteric antagonist (see details in our response to point 3).

In this section below, I ask out of interest as a reviewer, rather than suggesting that these be experimentally addressed, however, reference to these concepts should be touched on in the Discussion.

1- How do the authors think that the LFA-1 conformational change will be influenced when the T cells are not TCR stimulated, but rather via a CAR? What is happening inside the cell to promote the interplay between the CD3 stimulation and LFA-1.

Based on the work from the Jenkins lab (Davenport *et al.*, PNAS 2018), CAR-T cells form non-classical immune synapses with less frequent LFA-1 adhesion rings. It is tempting to speculate that the inside-out signaling emanating from the CAR is distinct from that emanating from the TCR. Based on the knowledge and analytical setups implemented in the current work, we believe that it would be very interesting indeed to compare LFA-1 activation following CAR vs TCR engagement. This perspective has been mentioned at the end of our discussion (line 405-407).

2- Similarly, while I really do understand the complexities of these experiments and using glass coverslips as targets for simplicity, have these experiments been done with TCR (like Tg OT-I) with diluted peptide presented on target cell membranes, in a cell-to-cell approach, and do authors see the same response or scaling LFA-1 open conformation with enhanced stimulation?

Murine transgenic T cells (OT-1 system) would be great but we lack a conformation-specific Ab for mouse high-affinity LFA-1. Human T cell clones with TCRs of selected specificity could be an alternative to the OKT3 Ab stimulation used here. We have added this in the discussion as a perspective to this work.

3- What role do the authors predict that CD8 is playing in the absence of MHCI. Have any of the experiments been conducted in the presence of anti-CD8 blocking antibody

To our knowledge the contribution of the CD8 co-receptor to the TCR-LFA-1 crosstalk has not been studied. The experimental setting we used in our study did not allow addressing this point that would require a proper TCR-pMHC1 system in which CD8 could be blocked, deleted or mutated.

4- The low affinity antibody Hi-111 is directly conjugated, however, the high affinity antibody m24 recognizing the 'open' conformation is biotin labelled and requires a secondary streptavidin binder, therefore increasing the stoichiometry of the labelling and improving the signal of the 'open' binder. For transparency this needs to be acknowledged and the intensity of labelling can not be directly compared, and authors should be careful to acknowledge this when quantitating sizes of clusters for example (Figure 1).

We are well aware that we cannot directly compare intensities for the 2 conformations since they were stained with antibodies coupled to different fluorochromes. We actually removed Supplementary Fig. 5 in which we had initially calculated the ratio of the 2 conformations.

In addition:

5- Small typos to be corrected:

Line 89: "specific of the" should read, Specific for the ...

Line 418 – Authors do not include the fluorophore for the Anti-CD107a antibody used (which they need to include) as this can severely impact the outcome, due to the quenching of pH instable fluorophores such as FITC.

The typos have been corrected. The fluorophore of our a-CD107a/LAMP1 antibody (PE) has been added line 471-472, such as the fluorochromes of the other antibodies of the experiment.

Supportive preliminary data:

Fig R1. ShRNA-mediated knock-down of ICAM-1 in target cells

A. ICAM-1 expression at the surface of P815 cells following transduction with ShRNA-encoding lentiviral vectors. **B.** LFA-1 conformation activation as detected with the m24 Ab staining, following stimulation of the CD8⁺ T cells with 4 versions of the P815 cells (WT, Mock-transduced, transduced with ICAM-1 ShRNA E and F) pre-coated with the indicated concentrations of anti-CD3 Ab. **C.** CD8⁺ T cell degranulation, as detected with the LAMP1 staining, following stimulation with 4 versions of the P815 cells (WT, Mock-transduced, transduced with ICAM-1 ShRNA E and F) pre-coated with the indicated concentrations of anti-CD3 Ab.

Fig R2. Analysis of the conformation of LFA-1 nanoclusters at the surface of CD8⁺ T cells

A. dSTORM imaging of the adhesion plane of a representative CD8⁺ T cells spreading on ICAM-1 and anti-CD3 Ab. LFA-1 was revealed with the conformation-independent TS2/4 antibody. Scale bar: 0.2 μm. **B.** Density (number of clusters per cell surface unit) of total (TS2/4), closed (HI111) and open (a24) LFA-1 nanoclusters, as quantified by dSTORM imaging in 4, 6 and 5 cells per staining, respectively. **C.** Confocal microscopy imaging of different LFA-1 conformations stained in the same cell (top: total LFA-1, middle: closed conformation, bottom: open conformation). Red arrows point to LFA-1 clusters that appear negative for both the closed and open conformation stainings.

REVIEWER COMMENTS

Reviewer #1 (Remarks to the Author):

The authors have addressed my concerns about a number of points, including the conservation of the closed state LFA-1, which is proposed to equilibrate over the entire cell surface, while the open state is trapped in the interface. That data is quite helpful. The analysis of the BIRT 377 effect and the quantitative effect of reducing ICAM-1 is also helpful. The authors have addressed my other comments.

Reviewer #2 (Remarks to the Author):

In the revised manuscript, the first of my concerns, i.e., that truly resting cells should be imaged rather than cells on PLL only, has been undertaken. This is helpful, as it suggests that contact with PLL is probably not introducing artefactual changes in LFA-1 organization.

However, the second of major concerns has not been adequately addressed, in my view. I had suggested that the authors show more rigorously that closed-state LFA-1 nanoclusters actually exist, rather than being "force-produced" by the algorithm. What I suggested would be helpful was that a STED analysis of, e.g., a membrane stain, or some other protein not thought to be clustered, needed be undertaken, so that it was clear that the much less "discretised" distribution of the Hi-111 Ab staining could be properly contextualised. It could well be that undulations in the surface topography of the cell could, e.g., explain the variation in staining that's observable in Fig. 1A. I still feel that this is a major point in terms of understanding the behaviour of LFA-1, i.e., whether it transitions from a freely diffusing state to a clustered one which would have been what I expected, or not. Simulations of randomness are helpful but not sufficient, as undulations in the membrane might also produce non-random distributions of wholly independent fluorescent species, such as membrane dyes.

My third concern was dealt with by the authors, by agreeing (also with Reviewer 1) that the observed dynamics of closed- and open-state changes upon activation was clearly incompatible with the digital conversion hypothesis.

Concern number four has been nicely dealt with through use of the BIRT 377 LFA-1 antagonist.

Finally, my concern over the very small effect of the anti-CD3 antibody on the ratio of closed/open state LFA-1, measured confocally in Fig. S5 of the original manuscript, has been dealt with by removing the data. I'm slightly uncomfortable about this. It would have been much better simply to have kept the data and explain that large changes in adhesion appear to be accompanied by relatively small changes in this ratio, or something like that. It seemed like a worthwhile analysis, just striking that it was so low, and relevant I think.

Reviewer #3 (Remarks to the Author):

Authors have addressed all points raised and have satisfied this reviewer

Point-by-point reply to reviewer 2

Our answers to the points raised by the reviewer appear in blue.
A figure to the attention of the reviewer (Fig. R2) has been added at the end of this document.

REVIEWER COMMENTS

Reviewer #2 (Remarks to the Author):

1) In the revised manuscript, the first of my concerns, i.e., that truly resting cells should be imaged rather than cells on PLL only, has been undertaken. This is helpful, as it suggests that contact with PLL is probably not introducing artefactual changes in LFA-1 organization.

We again thank the reviewer for this constructive criticism, which has inspired the staining of T cells in suspension as a way to capture LFA-1 organization independently from contacts with uncoated or PLL-coated glass.

2) However, the second of major concerns has not been adequately addressed, in my view. I had suggested that the authors show more rigorously that closed-state LFA-1 nanoclusters actually exist, rather than being "force-produced" by the algorithm. What I suggested would be helpful was that a STED analysis of, e.g., a membrane stain, or some other protein not thought to be clustered, needed be undertaken, so that it was clear that the much less "discretised" distribution of the Hi-111 Ab staining could be properly contextualised. It could well be that undulations in the surface topography of the cell could, e.g., explain the variation in staining that's observable in Fig. 1A. I still feel that this is a major point in terms of understanding the behaviour of LFA-1, i.e., whether it transitions from a freely diffusing state to a clustered one which would have been what I expected, or not. Simulations of randomness are helpful but not sufficient, as undulations in the membrane might also produce non-random distributions of wholly independent fluorescent species, such as membrane dyes.

In order to reinforce our initial STED analysis pointing to the fact that at least a proportion of closed-state LFA-1 might be assembled in nanoclusters, we had opted, in the first revision of the manuscript, for complementary *in silico* and experimental approaches (validation of the nanocluster segmentation algorithm with *in silico* generated images, probability estimations, application of dSTORM and SIM super resolution microscopy modalities).

We recognize that the reviewer's main suggestion was to perform the STED analysis with a membrane stain, or some other protein not thought to be clustered in order to contextualize the STED data obtained with the Hi-111 Ab staining. We initially considered that testing and analysing different candidate staining would take too much of the time/efforts we could possibly dedicate to the revision. In this second revision round, having now more time to dedicate to this question, we reconsidered this suggestion.

We first reasoned that a staining that would decorate multiple molecular entities at the cell surface might be a way to approach a random distribution. We therefore attempted to stain T cells spreading over ICAM-1 and anti-CD3 Abs with an AlexaFluor594-coupled Wheat germ agglutinin (WGA), a natural lectin that recognizes sugar chains (N-Acetylglucosamine residues and N-Acetylneuraminic acid) on various glycosylated receptors expressed by T cells. Live staining in conditions similar to those used for the Hi-111 Ab yielded a more

homogenous fluorescence distribution than that obtained with the Hi-111 Ab, at least for large areas of the synaptic plane (see **Figure R1** below). Other areas presented with more patchy patterns, possibly reflecting the non-uniform distribution of proteins such as the TCR/CD3 complex, CD43, CD44 and CD45 (see **Figure R1A** below).

We next reasoned that a dye with affinity for plasma membrane lipids should follow a process of free diffusion and might therefore be associated with a more uniform distribution over the entire synapse plane. We opted for testing MemBright 590 (idylle labs, Paris) a member of the MemBright membrane probe family, which inserts in the lipid bilayer via lipid anchors. Furthermore it has been shown to present limited internalization and to be adapted to super-resolution imaging¹. Staining of T cells spreading over ICAM-1 and anti-CD3 Abs with MemBright 590 yielded a fairly random fluorescence distribution over most of the T cell synaptic area (**Figure S9A**). Local fluctuations were clearly detected but they appeared to be on a reduced scale of intensity dispersion, as compared to the Hi-111 staining. Distinctly, the staining at the synapse edges presented a higher signal intensity and a dotted aspect. A possible interpretation for this edge effect is the enrichment of membrane related to the upper fold of the lamellipodial structure probably captured by STED due to limited Y axis resolution. Furthermore the edge of mature synapses is decorated with thin protrusions and folds associated with enriched content in membrane structures.

As a next step, we applied the cluster detection algorithm (WEKA segmentation model) to the MemBright 590 staining. In agreement with the enrichment of the staining at the edges, a majority of the detected events appeared to be detected at the edge of the synapse (see **Figure R1B-C**). We applied a mask to erode the cell edges in order to focus on the adhesive plane of the synapse. This process resulted in a drop of the density of clusters detected in cells stained with MemBright 590 but not in those stained with Hi-111 Ab, reflecting the fairly isotropic distribution of LFA-1 nanoclusters (see **Figure R1D**). Based on this analysis, we opted to show in the revised version of the paper the analysis of the synaptic plane after erosion of the edges (**Figure S9B**). The associated quantification clearly indicate that although the cluster detection algorithm recognized local enrichments of the MemBright 590 staining as clusters, the density of these events remained very low as compared to the Hi-111 Ab staining. We completed the new Figure S8 with the analysis of cluster detection on artificial images (**Figure S9D**), which indicates that our approach picks approximately 10 % of false-positive clusters from the background of randomly positioned molecules in addition to the LFA-1 clusters (**Figure S9E-F**). The new analysis presented in Figure S8 is explained in the method chapter "LFA-1 nanocluster segmentation algorithm" of the Method section.

To conclude, this new analysis is very useful in that it helps to contextualize the performance of the cluster detection algorithm. The fact that a higher density of clusters are detected with the Hi-111 Ab staining as compared to the MemGlow staining argues in favor of a non random assembly of closed LFA-1 molecules in clusters. On the other hand the fact that cluster-like events are detected with the MemGlow staining provides a direct assessment of the propensity of the algorithm to detect "false positives". This is in agreement with our complementary analysis of in silico generated images. Such events might correspond to local variations of lipid distribution at the plasma membrane. This is expected given the presence of lipid domains such as lipid rafts at the plasma membrane².

3) My third concern was dealt with by the authors, by agreeing (also with Reviewer 1) that the observed dynamics of closed- and open-state changes upon activation was clearly incompatible with the digital conversion hypothesis.

Although we have removed the concept of digital conversion from the title and have revised some of our data interpretations, we are discussing the possibility that such mechanism might operate. Indeed the apparent contradiction in the relative proportion of closed- and open-state clusters might come from the enrichment of closed conformation nanoclusters at

the synapse plane (as suggested by the flow cytometry data added as part of the first revision).

4) Concern number four has been nicely dealt with through use of the BIRT 377 LFA-1 antagonist.

We recognize that this point was indeed key. We have now confirmed it in multiple independent experiments.

5) Finally, my concern over the very small effect of the anti-CD3 antibody on the ratio of closed/open state LFA-1, measured confocally in Fig. S5 of the original manuscript, has been dealt with by removing the data. I'm slightly uncomfortable about this. It would have been much better simply to have kept the data and explain that large changes in adhesion appear to be accompanied by relatively small changes in this ratio, or something like that. It seemed like a worthwhile analysis, just striking that it was so low, and relevant I think.

We agree that the analysis of the ratio of closed/open state LFA-1 intensities is of interest. We have repositioned it as supplementary Figure. In line with the interpretation of the reviewer, we have commented on the fact that the LFA-1 conformation activation operates across a relatively limited range of values (in line with a limited proportion of clusters in the open state even upon strong activation) but that this translates into large changes in terms of adhesion.

References (have been added to the revised manuscript)

- 1 Collot M *et al.* MemBright: A Family of Fluorescent Membrane Probes for Advanced Cellular Imaging and Neuroscience. **Cell chemical biology** 26,600 (2019).
- 2 Saka SK *et al.* Multi-protein assemblies underlie the mesoscale organization of the plasma membrane. **Nat Commun** 5,4509 (2014).

Figure R1 (see next page)

Fig. R1. Additional data pertaining to the STED analysis of LFA-1 nanodistribution

A. STED image of the adhesion plane of a representative CD8⁺ T cell spreading on ICAM-1/anti-CD3 Ab for 20 min and stained with WGA. Scale bar: 5 μ m. **B.** STED images of the adhesion plane of representative CD8⁺ T cells spreading on ICAM-1/anti-CD3 Ab and staining with MemBright (up) or LFA-1 Hi-111 Ab (down). Scale bar: 5 μ m. Because of accumulation of MemBright staining at the edge of the cells, an erosion step (represented by the yellow line) was introduced to focus on the ventral adhesion place of the cells. **C.** LFA-1 clusters as detected by the trainable Weka segmentation algorithm applied to the STED image presented in A. The erosion step is represented by the yellow line. **D.** Density of clusters detected for the MemBright (n= 10 cells) and Hi-111 Ab (n= 40 cells) before and after application of the erosion step. Paired t-tests were applied, **** p<0.0001, * p<0.05.

REVIEWERS' COMMENTS

Reviewer #2 (Remarks to the Author):

The authors have done the experiment I asked for. The results are consistent with their claims. I don't see what purpose there is in masking-out the edges.

Point-by-point reply to reviewer 2

REVIEWER'S COMMENT

The authors have done the experiment I asked for. The results are consistent with their claims. I don't see what purpose there is in masking-out the edges.

We have better explained the masking-out of the edges in the legend of Supplementary Fig. 9b. For full transparency, we have replaced the images with masked edges by full images showing the delineation of the erosion mask (see new Supplementary Fig. 9b).